# A Novel Architecture for Integrating Shape Constraints in Neural Networks

## Abstract

This research proposes COMONet (Convex-Concave and Monotonicity-Constrained Neural Networks), a novel neural network architecture designed to embed inductive biases as shape constraints—specifically, monotonicity, convexity, concavity, and their combinations—into neural network training. Unlike previous models addressing only a subset of constraints, COMONet can comprehensively integrate and enforce eight distinct shape constraints: monotonic increasing, monotonic decreasing, convex, concave, convex increasing, convex decreasing, concave increasing, and concave decreasing. This integration is achieved through a unique partially connected structure, wherein inputs are grouped and selectively connected to specialized neural units employing either exponentiated or normal weights, combined with appropriate activation functions. Depending on the shape constraint required by each input, COMONet dynamically utilizes its full architecture or a partial configuration, providing significant flexibility. We further provide theoretical guarantees ensuring the strict enforcement of these constraints, while demonstrating that COMONet achieves performance comparable to existing benchmark methods. Moreover, our numerical experiments confirm that COMONet remains robust even under noisy conditions. Together, these results underscore COMONet's potential to advance constrained neural network training as a practical and theoretically grounded approach.

## 1 Introduction

Neural networks often struggle to align with domain knowledge when trained solely through error minimization, particularly when relying exclusively on observed data (Feelders, 2000; Dugas et al., 2009; Murdock et al., 2020). Domain knowledge refers to widely recognized or pre-established information specific to a given field (Yu et al., 2010; Muralidhar et al., 2018), and incorporating it into neural networks can enhance their reliability and interpretability. One effective approach to achieving this is through shape constraints, which encode well-defined relationships between input and output features (Groeneboom & Jongbloed, 2014; Johnson & Jiang, 2018). Ensuring that neural networks satisfy these constraints is particularly important in critical domains such as finance (Einav et al., 2013; Nelson et al., 2017), healthcare (Shahid et al., 2019), and law (Shahid et al., 2019), where accurate and reliable predictions are essential for informed decision-making and system optimization. As a result, there is growing interest in developing methods that integrate domain knowledge into neural network training, ensuring that learned models not only fit the data but also comply with real-world constraints and established principles. Among various possible constraints, monotonicity and convexity (or concavity) are two fundamental shape constraints that serve as inductive biases and are widely applied in several different domains (Amos et al., 2017; Kim & Lee, 2024). Monotonicity refers to a property where the output consistently non-decreases or non-increases[1] as the input increases. Meanwhile, convexity (or concavity) describes a function where, for any two points, the output does not exceed (does not fall below) the straight line connecting them, indicating an increasing (decreasing) rate of change. Fig. 1(a) illustrates various types of shape constraints related to monotonic increase (decrease) and convexity (or concavity) that can be incorporated into a model. It is important to note that monotonicity does not necessarily imply convexity or concavity, and convex or concave functions

---

[1]For readability, we refer to non-decreasing as increasing and non-increasing as decreasing throughout the rest of the paper.

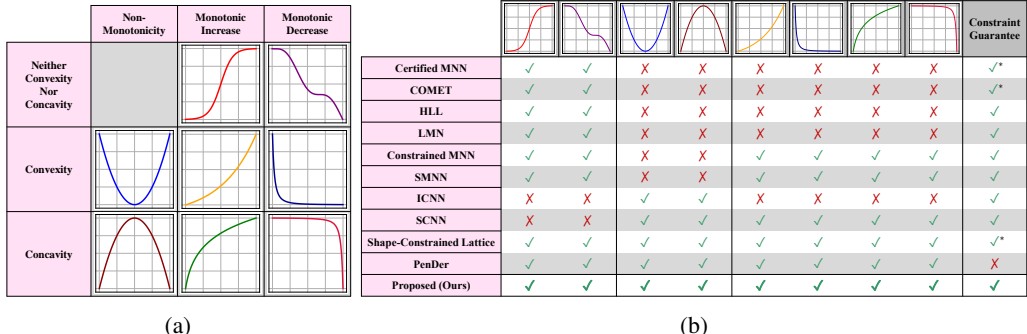

Figure 1: (a) Types of shape constraints. (b) Capability comparison with benchmark methods (∗ Indicates structural guarantees that may not hold in practice)

can be monotonically increasing, decreasing, or non-monotonic. Additionally, multiple distinct shape constraints can be independently and simultaneously imposed within a model.

Recent studies have shown growing interest in monotonic neural networks (Liu et al., 2020; Sivaraman et al., 2020; Runje & Shankaranarayana, 2023; Kim & Lee, 2024) and convex neural networks (Amos et al., 2017; Gupta et al., 2018), motivating various efforts to embed such properties into neural architectures. While substantial progress has been made in enforcing either monotonicity or convexity (or concavity) individually, research that jointly guarantees both types of constraints within a single model remains limited. Although several prior works attempt to incorporate multiple constraints, they often fail to cover all cases illustrated in Fig. 1(a) or to provide strict architectural guarantees. Mixed shape constraints—convexity, concavity, and monotonicity applied to different subsets of variables—naturally arise in engineered systems such as HVAC control (Zhang et al., 2017), concrete strength (Yeh, 2006), network resource allocation (Kelly et al., 1998), portfolio optimization (Markowitz, 2008), asset pricing (Gu et al., 2020; Fama & French, 2015; Breuer & Windisch, 2019) and physical dynamics (Goldstein et al., 1950; Spong et al., 2006; Armstrong-Hélouvry et al., 1994). These constraints are essential for safety, stability, and physical validity, and violations often lead to infeasible or unsafe behavior. In domains where strict adherence to shape constraints is critical, even small violations of monotonicity or convexity can result in unreliable predictions, reduced interpretability, and the loss of theoretical guarantees (Kim & Lee, 2024; Liu et al., 2020) or impact fairness considerations (Wang & Gupta, 2020). Fig. 1(b) compares existing methods, including ours, and highlights that most approaches cannot integrate all required shape behaviors when monotonicity and convexity (or concavity) must coexist.

To address these challenges, we propose COMONet (Convex-Concave and Monotonicity Constrained Neural Network), a novel yet simple neural network architecture designed to incorporate various shape constraints related to monotonicity and convexity within a single model. COMONet employs a partially connected structure, where input features are grouped and selectively connected to various types of specially designed units. Each unit utilizes either exponentiated or standard weights in combination with carefully chosen activation functions. This architecture enables the model to effectively learn diverse shape constraints while strictly enforcing all imposed constraints, thereby overcoming the limitations of existing approaches. Furthermore, depending on the types and composition of shape constraints that the entire set of input features must satisfy, COMONet can flexibly utilize either its full structure or only a partial configuration. This flexibility allows the model to enforce monotonicity, convexity (or concavity), or their combination as needed, ensuring strict compliance with the specified constraints.

## 2 RELATED WORK

**Monotonic neural networks:** Research on monotonic neural networks can be broadly categorized into two groups, regularization-based approaches and architecture-based approaches. The first group enforces monotonicity using various regularization techniques. For example, Certified MNN (Liu et al., 2020) applies penalties to partial derivatives, while COMET (Sivaraman et al., 2020) augments the dataset with so-called counter-examples for instances that violate monotonicity. These approaches have limitations, as they may not fully enforce monotonicity without strong regularization and often rely on external solvers such as MILP (Gurobi Optimization, LLC, 2023) and SMT (Barrett & Tinelli, 2018). The second group consists of hand-designed neural network architectures that inherently

guarantee monotonicity. Methods such as HLL (Yanagisawa et al., 2022), LMN (Nolte et al., 2022), Constrained MNN (Runje & Shankaranarayana, 2023), and SMNN (Kim & Lee, 2024) belong to this category. While some of these approaches are theoretically proven to ensure monotonicity, their restricted structures can lead to reduced predictive performance. As shown in Fig. 1(b), monotonic neural networks cannot naturally incorporate convexity or concavity, as they are explicitly designed to enforce monotonicity alone.

**Convex neural networks:** Convexity is a valuable property in model training, as it facilitates optimization, design, and control (Chen et al., 2018; Yang & Bequette, 2021). Due to these advantages, research on convex neural networks has gained significant interest. One of the earliest studies in this field introduced ICNN (Amos et al., 2017), which later inspired various extensions and modifications. For example, one extension leverages the difference between convex and concave components to approximate more complex functions (Sankaranarayanan & Rengaswamy, 2022). GON (Zhao et al., 2022) applied ICNN to optimization tasks, while FCNN (Pfrommer et al., 2024) was developed to enhance robustness against adversarial attacks. Additionally, a faster learning method for ICNN was proposed by introducing a novel initialization strategy (Hoedt & Klambauer, 2024). Expanding the concept of convexity, recent studies have explored monotonic-convexity, which refers to functions that are both monotonically increasing (decreasing) and convex (concave). An extension of convexity has also led to structures that combine monotonicity with convexity (or concavity), with SCNN (Gupta et al., 2018) being a representative example. However, SCNN can handle only convex or concave shapes and cannot model functions that are monotonic yet neither convex nor concave, such as $x + \sin(x)$. Moreover, it is unable to capture joint convex-concave interactions, where convex and concave variables coexist within the same function.

**Shape constrained neural networks:** Among existing approaches, PenDer (Gupta et al., 2021) and Shape-Constrained Lattice models (SCL) (Gupta et al., 2018) most closely support the full set of constraints in Fig. 1(a), but both have structural limitations. PenDer uses regularization, encouraging but not guaranteeing constraint satisfaction; violations must be detected post hoc rather than prevented. SCL enforces convexity, concavity, and monotonicity through a discretized lattice parameterization. Although the lattice structure can theoretically satisfy these constraints, its resolution induces a trade-off between computational cost and approximation fidelity. To ease this burden, recent work trains multiple low-dimensional lattices over randomly selected feature subsets and combines them via an ensemble (Milani Fard et al., 2016; Gupta et al., 2016). Training further requires projected gradient updates over a large number of linear inequality constraints, which grows rapidly with the lattice resolution. SCL implementations therefore rely on stochastic constraint sampling (Cotter et al., 2016), projecting only a small subset at each iteration. Constraint satisfaction may not be guaranteed at every training step, and temporary violations can occur before convergence. Moreover, lattice interpolation supports per-feature constraints but cannot model joint convex or joint concave curvature, which is essential in many optimization or curvature-sensitive tasks (Gupta et al., 2018). These limitations motivate a unified, end-to-end differentiable framework that guarantees all shape constraints without approximate projections or post-hoc verification.

## 3 SHAPE CONSTRAINTS

We consider a continuous, differentiable multivariate function $f: [0,1]^d \to \mathbb{R}$. We consider three classes of local shape constraints—*partial (joint) convexity*, *partial (joint) concavity* and *partial monotonicity*—that apply to subsets of the input coordinates. Let $[d] = \{1, \ldots, d\}$ and we denote vectors in $\mathbb{R}^d$ by bold lowercase letters (e.g., $\mathbf{x}$, $\mathbf{t}$) and index-sets by calligraphic uppercase letters (e.g., $\mathcal{CV}$, $\mathcal{MN}$). To impose distinct shape constraints on different input dimensions, we partition coordinates of $\mathbf{x} \in \mathbb{R}^d$ into six disjoint groups:

$$\mathbf{x} = \left(\mathbf{x}_{cv}, \mathbf{x}_{mv}, \mathbf{x}_{cc}, \mathbf{x}_{mc}, \mathbf{x}_{mn}, \mathbf{x}_u\right) \in \mathbb{R}^{|\mathcal{CV}|} \times \mathbb{R}^{|\mathcal{MV}|} \times \mathbb{R}^{|\mathcal{CC}|} \times \mathbb{R}^{|\mathcal{MC}|} \times \mathbb{R}^{|\mathcal{MN}|} \times \mathbb{R}^{|\mathcal{U}|},$$

where, $\mathcal{CV} \cup \mathcal{MV} \cup \mathcal{CC} \cup \mathcal{MC} \cup \mathcal{MN} \cup \mathcal{U} = [d]$. Each index-set enforces a particular constraint on $f$: $\mathcal{CV}$ (convex only), $\mathcal{MV}$ (monotonic + convex), $\mathcal{CC}$ (concave only), $\mathcal{MC}$ (monotonic + concave), $\mathcal{MN}$ (monotonic only), and $\mathcal{U}$ (unconstrained). Further, $\mathcal{V} = \mathcal{CV} \cup \mathcal{MV}$, $\mathcal{C} = \mathcal{CC} \cup \mathcal{MC}$, $\mathcal{M} = \mathcal{MN} \cup \mathcal{MV} \cup \mathcal{MC}$. represents the set contains each, all convex, concave and monotonic features. **Partial convexity and partial joint convexity:** Partition $\mathbf{x} = (\mathbf{x}_v, \mathbf{x}_{\neg v}) \in \mathbb{R}^d$ with $\mathbf{x}_v \in \mathbb{R}^{|\mathcal{V}|}$ and $\mathbf{x}_{\neg v} \in \mathbb{R}^{d-|\mathcal{V}|}$. For each coordinate $i \in \mathcal{V}$, write $\mathbf{x} = (x_i, \mathbf{x}_{\neg i})$, where $\mathbf{x}_{\neg i}$ denotes all coordinates except $x_i$. We say that $f$ is *partially convex* in coordinate $x_i$ iff, for any fixed $\mathbf{x}_{\neg i}$, any $x_i, x_i' \in \mathbb{R}$, and any $\lambda \in [0,1]$, $f(\lambda x_i + (1 - \lambda)x_i', \mathbf{x}_{\neg i}) \leq \lambda f(x_i, \mathbf{x}_{\neg i}) + (1 - \lambda) f(x_i', \mathbf{x}_{\neg i})$. Further, we

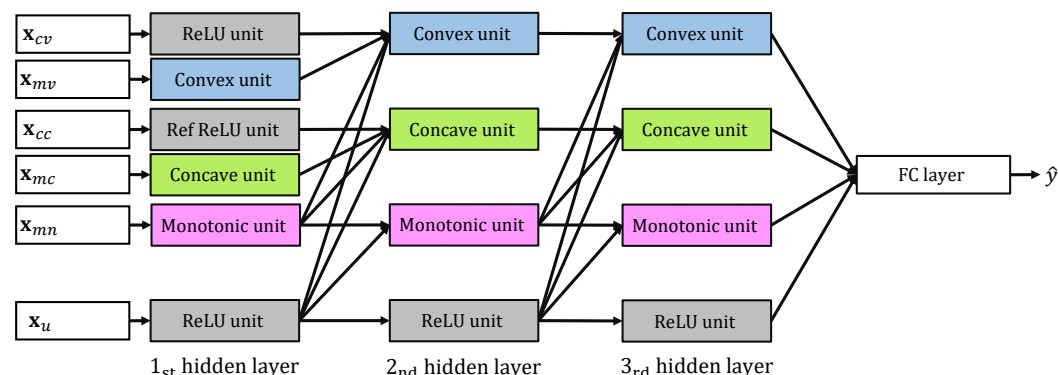

Figure 2: Structure of COMONet that has 3 hidden layers.

say $f$ is *partially joint convex* on $\mathbf{x}_v$ iff, for any fixed $\mathbf{x}_{\neg v}$ and all $\mathbf{x}_v, \mathbf{x}'_v \in \mathbb{R}^{|\mathcal{V}|}$ and $\lambda \in [0, 1]$,
$$f\left(\lambda \mathbf{x}_v + (1 - \lambda) \mathbf{x}'_v, \mathbf{x}_{\neg v}\right) \leq \lambda f(\mathbf{x}_v, \mathbf{x}_{\neg v}) + (1 - \lambda) f(\mathbf{x}'_v, \mathbf{x}_{\neg v}).$$

**Partial concavity and partial joint concavity.** Partition $\mathbf{x} = (\mathbf{x}_c, \mathbf{x}_{\neg c}) \in \mathbb{R}^d$ with $\mathbf{x}_c \in \mathbb{R}^{|\mathcal{C}|}$ and $\mathbf{x}_{\neg c} \in \mathbb{R}^{d - |\mathcal{C}|}$. For each coordinate $i \in \mathcal{C}$, write $\mathbf{x} = (x_i, \mathbf{x}_{\neg i})$, where $\mathbf{x}_{\neg i}$ denotes all coordinates except $x_i$. We say that $f$ is *partially concave* in coordinate $x_i$ iff, for any fixed $\mathbf{x}_{\neg i}$, any $x_i, x'_i \in \mathbb{R}$, and any $\lambda \in [0, 1]$, $f(\lambda x_i + (1 - \lambda) x'_i, \mathbf{x}_{\neg i}) \geq \lambda f(x_i, \mathbf{x}_{\neg i}) + (1 - \lambda) f(x'_i, \mathbf{x}_{\neg i})$. Further, we say $f$ is *partially joint concave* on $\mathbf{x}_c$ iff, for any fixed $\mathbf{x}_{\neg c}$, any $\mathbf{x}_c, \mathbf{x}'_c \in \mathbb{R}^{|\mathcal{C}|}$, and any $\lambda \in [0, 1]$,
$$f\left(\lambda \mathbf{x}_c + (1 - \lambda) \mathbf{x}'_c, \mathbf{x}_{\neg c}\right) \geq \lambda f(\mathbf{x}_c, \mathbf{x}_{\neg c}) + (1 - \lambda) f(\mathbf{x}'_c, \mathbf{x}_{\neg c}).$$

**Partial monotonicity:** Partition $\mathbf{x} = (\mathbf{x}_m, \mathbf{x}_{\neg m}) \in \mathbb{R}^d$ with $\mathbf{x}_m \in \mathbb{R}^{|\mathcal{M}|}$, $\mathbf{x}_{\neg m} \in \mathbb{R}^{d - |\mathcal{M}|}$. We say $f$ is *partially monotonic increasing* on $\mathbf{x}_m$ iff, $\frac{\partial f}{\partial x_i} \geq 0, \forall i \in |\mathcal{M}|$. And, we say $f$ is *partially monotonic decreasing* on $\mathbf{x}_m$ iff, $\frac{\partial f}{\partial x_i} \leq 0, \forall i \in |\mathcal{M}|$. (For monotonicity, enforcing monotonicity on each variable individually guarantees joint monotonicity.)

By combining the definitions of partial monotonicity (increasing or decreasing) with partial convexity (or concavity), we can specify all eight shape-constraint types depicted in Fig. 1(a).

## 4 PROPOSED METHOD

**Fundamental units of COMONet:** A key aspect of our approach is employing five distinct unit types to effectively integrate and enforce diverse shape constraints. These units incorporate either exponentiated or conventional weights, using ReLU or capped ReLU-$n$ (Liew et al., 2016) as activation functions. For any real scalar $z$, we define ReLU $= (z)_+ = \max(0, z)$ and ReLU-$n = (z)_+^n = \min\{n, \max(0, z)\}$. The equations below define the five units, each of which takes the vector $\mathbf{t}$ as input:

$$convex\ unit \coloneqq h_{\text{conv}}(\mathbf{t}) = \left(\exp(\mathbf{W})^\top \mathbf{t} + \mathbf{b}\right)_+ \tag{1}$$

$$concave\ unit \coloneqq h_{\text{conc}}(\mathbf{t}) = -\left(-\left(\exp(\mathbf{W})^\top \mathbf{t} + \mathbf{b}\right)\right)_+ \tag{2}$$

$$monotonic\ unit \coloneqq h_{\text{mono}}(\mathbf{t}) = \left(\exp(\mathbf{W})^\top \mathbf{t} + \mathbf{b}\right)_+^n \tag{3}$$

$$relu\ unit \coloneqq h_{\text{relu}}(\mathbf{t}) = \left(\mathbf{W}^\top \mathbf{t} + \mathbf{b}\right)_+ \tag{4}$$

$$ref\text{-}relu\ unit \coloneqq h_{\text{ref-relu}}(\mathbf{t}) = -\left(-\left(\mathbf{W}^\top \mathbf{t} + \mathbf{b}\right)\right)_+ \tag{5}$$

$h_{\text{conv}}$, $h_{\text{conc}}$ and $h_{\text{mono}}$ utilize exponentiated weights (Zhang & Zhang, 1999; Agarwal et al., 2021; Dinh et al., 2016) to constraint reparametrized weight to be positive. $h_{\text{mono}}$ employs $(z)_+^n$ which contains both convex and concave hinge components. Further, $h_{\text{conc}}$ and $h_{\text{ref-relu}}$ employ point-symmetric variants of ReLU, $-(-(z)_+)$ and ReLU-$n$, $-(-(z)_+^n)$. We adopt ReLU and ReLU-$n$ by default, using ReLU as our baseline for its computational efficiency, resilience to vanishing gradients, and piecewise linear sparsity that accelerates convergence and boosts generalization (Nair & Hinton, 2010; Glorot et al., 2011). As shown in Appendix G.1, any activation functions meeting the required

characteristics can be used instead. Each fundamental unit defined above satisfies the following properties, as formalized in the lemmas below:

**Lemma 4.1.** *Let $h_{\text{conv}} : \mathbb{R}^d \to \mathbb{R}^k$ and denote its $j$th coordinate by $f_j(\mathbf{t}) := \left[h_{\text{conv}}(\mathbf{t})\right]_j = \left(\exp(\mathbf{w}_j)^\top \mathbf{t} + b_j\right)_+$. Then, $\forall j \in [k]$, $f_j$ is jointly convex in $\mathbf{t}$ and $f_j$ is coordinatewise increasing, i.e. $\frac{\partial f_j}{\partial t_i} \geq 0$, $\forall i \in [d]$.*

**Lemma 4.2.** *Let $h_{\text{relu}} : \mathbb{R}^d \to \mathbb{R}^k$ and denote its $j$th coordinate by $f_j(\mathbf{t}) := \left[h_{\text{relu}}(\mathbf{t})\right]_j = \left(\mathbf{w}_j^\top \mathbf{t} + b_j\right)_+$. Then, $\forall j \in [k]$, $f_j$ is jointly convex in $\mathbf{t}$.*

**Lemma 4.3.** *Let $h_{\text{conc}} : \mathbb{R}^d \to \mathbb{R}^k$ and denote its $j$th coordinate by $f_j(\mathbf{t}) := \left[h_{\text{conc}}(\mathbf{t})\right]_j = -\left(-\exp(\mathbf{w}_j)^\top \mathbf{t} - b_j\right)_+$. Then, $\forall j \in [k]$, $f_j$ is jointly concave in $\mathbf{t}$ and coordinatewise increasing, i.e. $\frac{\partial f_j}{\partial t_i} \geq 0$, $\forall i \in [d]$.*

**Lemma 4.4.** *Let $h_{\text{ref-relu}} : \mathbb{R}^d \to \mathbb{R}^k$ and denote its $j$th coordinate by $f_j(\mathbf{t}) := \left[h_{\text{ref-relu}}(\mathbf{t})\right]_j = -\left(-\mathbf{w}_j^\top \mathbf{t} - b_j\right)_+$. Then, $\forall j \in [k]$, $f_j$ is jointly concave in $\mathbf{t}$.*

**Lemma 4.5.** *Let $h_{\text{mono}} : \mathbb{R}^d \to \mathbb{R}^k$ and denote its $j$th coordinate by $f_j(\mathbf{t}) := \left[h_{\text{mono}}(\mathbf{t})\right]_j = \left(\exp(\mathbf{w}_j)^\top \mathbf{t} + b_j\right)_+^n$. Then, $\forall j \in [k]$, $f_j$ is coordinatewise increasing in $\mathbf{t}$, i.e. $\frac{\partial f_j}{\partial t_i} \geq 0$, $\forall i \in [d]$.*

Detailed proofs for lemma 4.3-4.5 are provided at Appendix A.

**Network structure:** The proposed architecture, illustrated in Fig. 2, adopts a selectively connected design that routes information through designated subsets of connections. This structure is conceptually similar to the architectures proposed in (Amos et al., 2017) and (Kim & Lee, 2024), which also utilize partially connected designs. Such a design ensures that the specific properties of individual input variable groups are preserved, while simultaneously allowing the model to effectively capture the interactions among all input variables. Let $h^{(i)}$ be the $i$-th hidden layer. Then the overall formulation of COMONet, which employs the five distinct types of units with depth $l$ defined above, is presented below:

When $i = 1$,

$$h^{(1)} = \left[ h_{\text{relu},cv}^{(1)}(\mathbf{x}_{cv}), h_{\text{conv}}^{(1)}(\mathbf{x}_{mv}), h_{\text{ref-relu}}^{(1)}(\mathbf{x}_{cc}), h_{\text{conc}}^{(1)}(\mathbf{x}_{mc}), h_{\text{mono}}^{(1)}(\mathbf{x}_{mn}), h_{\text{relu},u}^{(1)}(\mathbf{x}_u) \right]. \quad (6)$$

When $i = 2$,

$$h_{\text{conv}}^{(2)} = h_{\text{conv}}([h_{\text{relu},cv}^{(1)}(\mathbf{x}_{cv}), h_{\text{conv}}^{(1)}(\mathbf{x}_{mv}), h_{\text{mono}}^{(1)}(\mathbf{x}_{mn}), h_{\text{relu},u}^{(1)}(\mathbf{x}_u)]) \quad (7)$$

$$h_{\text{conc}}^{(2)} = h_{\text{conc}}([h_{\text{ref-relu}}^{(1)}(\mathbf{x}_{cc}), h_{\text{conc}}^{(1)}(\mathbf{x}_{mc}), h_{\text{mono}}^{(1)}(\mathbf{x}_{mn}), h_{\text{relu},u}^{(1)}(\mathbf{x}_u)]) \quad (8)$$

$$h_{\text{mono}}^{(2)} = h_{\text{mono}}([h_{\text{mono}}^{(1)}(\mathbf{x}_{mn}), h_{\text{relu},u}^{(1)}(\mathbf{x}_u)]) \quad (9)$$

$$h_{\text{relu}}^{(2)} = h_{\text{relu}}([h_{\text{relu},u}^{(1)}(\mathbf{x}_u)]) \quad (10)$$

$$h^{(2)} = \left[ h_{\text{conv}}^{(2)}, h_{\text{conc}}^{(2)}, h_{\text{mono}}^{(2)}, h_{\text{relu}}^{(2)} \right] \quad (11)$$

When $i \geq 3$,

$$h_{\text{conv}}^{(i)} = h_{\text{conv}}([h_{\text{conv}}^{(i-1)}, h_{\text{mono}}^{(i-1)}, h_{\text{relu}}^{(i-1)}]) \quad (12)$$

$$h_{\text{conc}}^{(i)} = h_{\text{conc}}([h_{\text{conc}}^{(i-1)}, h_{\text{mono}}^{(i-1)}, h_{\text{relu}}^{(i-1)}]) \quad (13)$$

$$h_{\text{mono}}^{(i)} = h_{\text{mono}}([h_{\text{mono}}^{(i-1)}, h_{\text{relu}}^{(i-1)}]) \quad (14)$$

$$h_{\text{relu}}^{(i)} = h_{\text{relu}}([h_{\text{relu}}^{(i-1)}]) \quad (15)$$

$$h^{(i)} = \left[ h_{\text{conv}}^{(i)}, h_{\text{conc}}^{(i)}, h_{\text{mono}}^{(i)}, h_{\text{relu}}^{(i)} \right] \quad (16)$$

Let $h^{(l)}$ the output vector of $l$th hidden layer, and $f(\mathbf{x})$ be the output node in a fully connected output layer, then,

$$f(\mathbf{x}) = \exp\left(\mathbf{W}\right)^\top h^{(l)} + \mathbf{b} \quad (17)$$

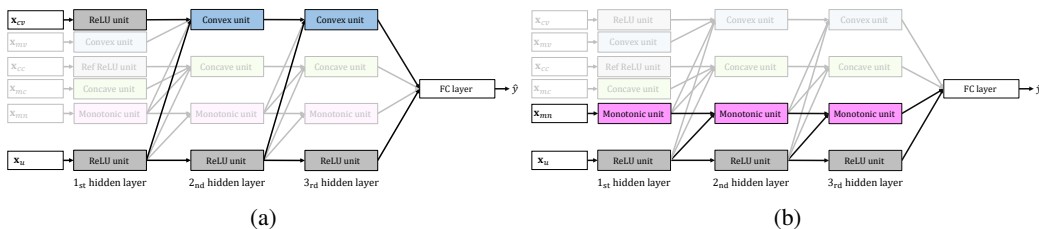

Figure 3: **Configuration examples:** (a) Configuration of COMONet when the input consists only of $\mathbf{x}_{cv}$ and $\mathbf{x}_u$, (b) Configuration of COMONet when the input consists only of $\mathbf{x}_{mn}$ and $\mathbf{x}_u$.

Where, $\exp(\mathbf{W})$ and $\mathbf{b}$ are the exponentiated weight matrix and bias vector between $l$th hidden layer and output layer. Although we present the formulation with a single output node, it naturally extends to multiple output nodes without issue. The above formulation enables COMONet to train a neural network that enforces the specified constraints on each variable. Monotonically decreasing features in $\mathcal{MV}, \mathcal{MC}, \mathcal{MN}$ are multiplied by –1 before training—transforming them into increasing inputs—and their original sign is restored at inference, enabling seamless integration with other shape constraints.

**Shape-constraint guarantee of COMONet:**  Following the definitions in Section 3, we now demonstrate that COMONet satisfies the convexity, concavity, and monotonicity properties. The proofs of these theorems proceed by invoking the lemma 4.3-4.5 that characterize each unit's properties.

**Theorem 4.6** (Convexity of COMONet). *Let $f(\mathbf{x})$ be the proposed COMONet, which has $l$ hidden layers. Partition the input $\mathbf{x} \in \mathbb{R}^d$ as $\mathbf{x} = (\mathbf{x}_v, \mathbf{x}_{\neg v})$, $\mathbf{x}_v = \{x_i \mid i \in \mathcal{V}\}, \mathcal{V} \subseteq [d]$. Then $f(\mathbf{x})$ is* partially jointly convex *with respect to* $\mathbf{x}_v$.

**Theorem 4.7** (Concavity of COMONet). *Let $f(\mathbf{x})$ be the proposed COMONet, which has $l$ hidden layers. Partition the input $\mathbf{x} \in \mathbb{R}^d$ as $\mathbf{x} = (\mathbf{x}_c, \mathbf{x}_{\neg c})$, $\mathbf{x}_c = \{x_i \mid i \in \mathcal{C}\}, \mathcal{C} \subseteq [d]$. Then $f(\mathbf{x})$ is* partially jointly concave *with respect to* $\mathbf{x}_c$.

**Theorem 4.8** (Monotonicity of COMONet). *Let $f(\mathbf{x})$ be the proposed COMONet, which has $l$ hidden layers. Partition the input $\mathbf{x} \in \mathbb{R}^d$ as $\mathbf{x} = (\mathbf{x}_m, \mathbf{x}_{\neg m})$, $\mathbf{x}_m = \{x_i \mid i \in \mathcal{M}\}, \mathcal{M} \subseteq [d]$. Then $f(\mathbf{x})$ is* partially monotonic increasing *with respect to* $\mathbf{x}_m$. *In particular, for each $x_i$ with $i \in \mathcal{M}$, $f$ is monotonic (increasing) in $x_i$.*

Detailed proofs for Theorem 4.6-4.8 are provided at Appendix A. Flow diagrams for each variable group appear in Appendix A. As illustrated in Appendix B, the unconstrained features $\mathbf{x}_u$ are processed by multiple standard ReLU layers—without any shape constraints—allowing them to fully exploit their expressive capacity as they propagate through the network. Moreover, at each hidden layer, $\mathbf{x}_u$'s activations are routed into the convex, concave, and monotonic units, enabling it to interact with all other variable groups. Finally, Appendix G.2 demonstrates the overall effectiveness of the proposed network structure.

**High flexibility and modularity:**  Proposed model demonstrates high modularity and flexibility, enabling it to be easily tailored to accommodate various relationships and properties of input variables. This adaptability stems from the structural characteristics of the proposed method, which employs a partially connected structure. For instance, when the only constraint is convexity—i.e. we partition the input as $\mathbf{x} = (\mathbf{x}_{cv}, \mathbf{x}_u) \in \mathbb{R}^{|\mathcal{CV}|} \times \mathbb{R}^{|\mathcal{U}|}$. In this case, the resulting configuration—shown in Fig. 3 (a)—closely resembles the PICNN (Amos et al., 2017) architecture. Similarly, when the only constraint is monotonicity—i.e. we partition the input as $\mathbf{x} = (\mathbf{x}_{mn}, \mathbf{x}_u) \in \mathbb{R}^{|\mathcal{MN}|} \times \mathbb{R}^{|\mathcal{U}|}$, the resulting configuration—shown in Fig. 3(b)—aligns with the SMNN (Kim & Lee, 2024) architecture.

**Interaction layer:** To enhance the expressive power of COMONet while strictly preserving variable-wise convexity and concavity constraints, we introduce an optional interaction layer.

$$\mathcal{I}_{\text{cross}}(\mathbf{x}) = \sum_{i \in \mathcal{CC}} \sum_{j \in \mathcal{CV}} \alpha_{i,j}\, x_i x_j, \tag{18}$$

$$\mathcal{I}_{\text{intra}}(\mathbf{x}) = \sum_{i \in \mathcal{CV},\, j \in \mathcal{CV},\, i \neq j} \beta_{i,j}\, x_i x_j + \sum_{i \in \mathcal{CC},\, j \in \mathcal{CC},\, i \neq j} \beta_{i,j}\, x_i x_j. \tag{19}$$

First, cross-group interactions in equation 18 enable communication between $\mathcal{CV}$ and $\mathcal{CC}$, which are otherwise processed independently in COMONet. Introducing such interactions is essential for enhancing the representational capacity of the model. Second, Intra-group interactions in equation 19 enable interactions within $\mathcal{CV}$ and $\mathcal{CC}$. Although COMONet inherently imposes strong global curvature constraints such as joint convexity or joint concavity, some tasks benefit from coordinate-wise curvature constraints (*ceteris paribus*), and intra-group interactions provide a mechanism for this relaxation. A key mathematical requirement for any interaction function $\mathcal{I}$ is that it must not modify the curvature assigned to each variable; Accordingly, its second-order partial derivatives with respect to each involved variable must identically vanish, ensuring that no sign change in curvature can occur under any input.

**Theorem 4.9.** *Let $f : \mathbb{R}^2 \to \mathbb{R}$ be twice differentiable and satisfy: for all $x_2 \in \mathbb{R}$, the mapping $x_1 \mapsto f(x_1, x_2)$ is convex, and for all $x_1 \in \mathbb{R}$, the mapping $x_2 \mapsto f(x_1, x_2)$ is concave. Then, if we decompose*

$$f(x_1, x_2) = g(x_1) + h(x_2) + \phi(x_1, x_2),$$

*where $g$ depends only on $x_1$ and $h$ depends only on $x_2$, then the pure interaction term $\phi$, which preserves the convex–concave assignments for any admissible choices of $g$ and $h$, must be*

$$\phi(x_1, x_2) = \alpha\, x_1 x_2, \qquad \alpha \in \mathbb{R}.$$

**Theorem 4.10.** *Let $f : \mathbb{R}^2 \to \mathbb{R}$ be twice differentiable and jointly convex (or jointly concave) in $(x_1, x_2)$, and assume that $x_1$ and $x_2$ belong to the same curvature group of COMONet (both $\mathcal{CV}$ or both $\mathcal{CC}$). Consider adding the bilinear term*

$$\phi(x_1, x_2) = \beta\, x_1 x_2, \qquad \beta \in \mathbb{R},$$

*and redefine the mapping by*

$$f(x_1, x_2) := f(x_1, x_2) + \phi(x_1, x_2).$$

*Then this addition preserves the assigned per-variable constraints while allowing the joint convexity (or concavity) in $(x_1, x_2)$ to be relaxed.*

Theorem 4.10 further establishes that such bilinear interactions preserve the constraints of each variable while allowing controlled relaxation from joint curvature to separate curvature (*ceteris paribus*). The proofs of Theorem 4.9 and Theorem 4.10 are provided in Appendix A, and further implementation details of the interaction layer appear in Appendix C.

## 5 NUMERICAL EXPERIMENTS

### 5.1 EXPERIMENTS ON SYNTHETIC DATASETS

We first evaluated the effectiveness of our proposed method through experiments on synthetic datasets, aiming to demonstrate that COMONet can satisfy various shape constraints. Since our method is theoretically proven to guarantee these constraints and has been shown to be adaptable to different types of inputs, testing all possible combinations of shape constraints introduced in Section 1 would be unnecessary. Instead, we conducted experiments on two synthetic datasets where different types of inputs were appropriately mixed and presented the results.

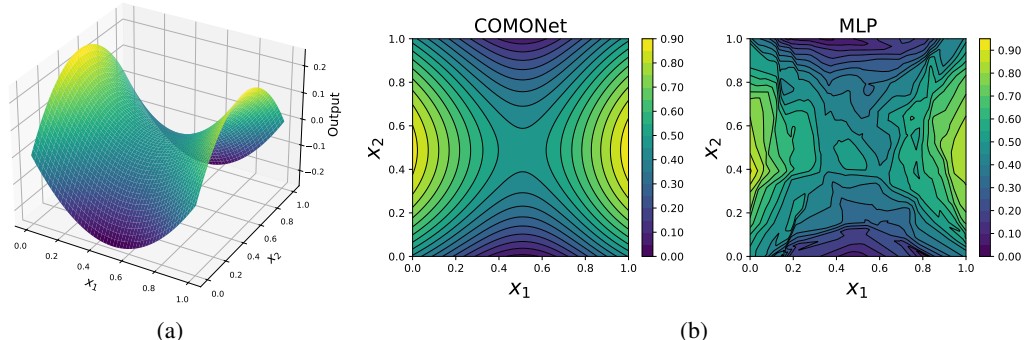

Figure 4: (a) Hyperbolic Paraboloid function (b) Contour plots of the fitted models by COMONet (Left) and MLP (Right) with respect to $x_1$ and $x_2$ when $\lambda = 0.05$

**Convexity and Concavity:**  The well-known hyperbolic paraboloid shown in equation 20 and Fig. 4(a) was chosen because it exhibits both convexity and concavity. In equation 20, $x_1$ is a convex input, while $x_2$ is a concave input of $y$. To the best of our knowledge, with the exception of the Pender method, previous related studies have not addressed cases in which both convexity and concavity must be satisfied simultaneously. To demonstrate that COMONet accurately fits the function even in noisy environments, we introduced Gaussian noise $\epsilon$ and varied the noise level parameter $\lambda$.

$$y = (x_1 - 0.5)^2 - (x_2 - 0.5)^2 + \lambda\epsilon, \quad \epsilon \sim N(0, 1), x_i \in [0, 1], i = 1, 2, \quad \lambda \in \{0, 0.05, 0.1, 0.2\}. \tag{20}$$

Since the inputs are exclusively convex and concave, we utilized the COMONet structure shown in Appendix E.5. for this experiment. Specifically, the ReLU layer and the convex layer were applied to $x_1$, while the reflected ReLU layer and the concave layer were used for $x_2$. For comparison, a traditional MLP was employed as the baseline method. At each noise level, we generated 1,000 instances and split them into training (80%) and test (20%) sets.

The test mean squared errors (MSEs) are presented in Table 1. As shown in Table 1, MLP outperformed COMONET in the absence of noise. However, as the noise level increased, COMONet demonstrated better performance, with a smaller increase in MSE compared to MLP. This demonstrates that COMONet provides a robust fit to the function. More importantly,

Table 1: Test MSEs of COMONet and MLP

| $\lambda$ | 0 | 0.05 | 0.1 | 0.2 |
|---|---|---|---|---|
| COMONet | 0.0010 | 0.0028 | 0.0117 | 0.0448 |
| MLP | 0.0001 | 0.0030 | 0.0124 | 0.0529 |

consider Fig. 4(b), which displays the contour plots of the fitted models from both methods. Even with a small amount of noise, MLP failed to preserve convexity and concavity, whereas our method consistently maintained these constraints. Although we omitted further visualizations, our method continues to satisfy them as noise levels increase.

**Monotonicity and Convexity:**  We extended our experiment to a case with more shape constraints. Specifically, we designed a 4-dimensional example, with the function defined in equation 21.

$$y = \frac{2\pi x_1 + \sin(2\pi x_1)}{2\pi} + (x_2 - 0.5)^2 + e^{x_3} + \cos(2\pi x_4) + \lambda\epsilon, \quad \epsilon \sim N(0, 1). \tag{21}$$
$$x_i \in [0, 1], i = 1, 2, 3, 4, \quad \lambda \in \{0, 1, 2, 5, 10, 20\}.$$

As shown in equation 21, a distinct shape constraint was assigned to each input feature: $x_1$ is a monotonically increasing feature, $x_2$ is convex, $x_3$ is monotonic-convex, and $x_4$ is unconstrained in the noiseless setting. Similar to the previous experiment, we introduced Gaussian noise and controlled the noise level by adjusting the $\lambda$ value. At each noise level, we generated 3,000 instances and split them into training (80%) and test (20%) sets. For comparison, two baseline methods were used in this experiment. In addition to the traditional MLP, we included a model referred to as "Same Structure" (shown in Appendix F.1) which shares the same architecture as COMONet but replaces all units with ReLU layers, meaning no shape constraints were enforced. To ensure a fair comparison, all models were constructed with an identical number of nodes per unit. The performance results in terms of MSE are presented in Fig. 5(a). The results indicate that MLP achieved the lowest training MSE

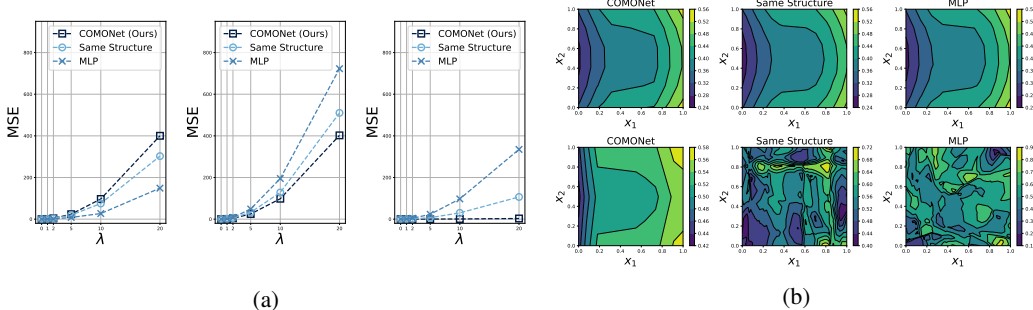

(a)                                                            (b)

Figure 5: (a) MSEs from the experiments at different $\lambda$ values. (left) Training MSEs, (middle) Test MSEs, (right) Denoised Test MSEs. (b) Contour plots in the $x_1$-$x_2$ plane for $\lambda = 0$ (top) and $\lambda = 1$ (bottom), comparing outputs of COMONet, same structure, and MLP.

across all $\lambda$ values, followed by the Same Structure model, while COMONet exhibited the highest training error. In contrast, for test MSE, COMONet consistently outperformed the other models, while MLP exhibited the highest error. This indicates that MLP overfits the noise during training, whereas our method does not. Notably, in the denoised test MSE—which evaluates performance in predicting the noise-free ground truth—COMONet maintained robust predictions even as $\lambda$ increased. These results quantitatively confirm that the shape constraints enforced by COMONet not only mitigate the impact of noise but also significantly improve the model's generalization performance.

Fig. 5(b) shows the contour plots of the fitted models for all three approaches. In the absence of noise, all models produced reasonable fits. However, even with a small noise level ($\lambda = 1$), the MLP and Same Structure models completely failed to satisfy the constraints, whereas COMONet successfully preserved monotonicity with respect to $x_1$ and convexity with respect to $x_2$. This figure confirms that our method effectively maintains the imposed shape constraints.

**Trustworthy test using LIME:** Fig. 6 demonstrates that the proposed method prevents incorrect interpretations. It shows the LIME (Ribeiro et al., 2016) values for $x_1$ and $x_3$ at the different noise levels. Notice that $x_1$ is a monotonically increasing feature and $x_3$ is monotonic-convex, meaning their LIME values should always be positive for a correct interpretation. As shown in the figure, the LIME values for both $x_1$ and $x_3$ computed from the COMONet models are consistently positive, with low variance, which aligns with the expected interpretation. By contrast, LIME values from the MLP models fluctuate in sign and exhibit high variance. These results validate that embedding shape constraints yields more reliable interpretations.

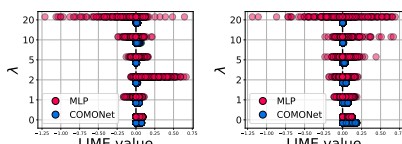

Figure 6: LIME values for $x_1$ (Left) and $x_3$ (Right), with data sampled at $x_2$ and $x_4$ fixed at their mean values in the test dataset. The values are shown for COMONet (red) and MLP (blue) across different $\lambda$ values.

## 5.2 EXPERIMENTS ON REAL-WORLD DATASETS

We now present the general performance level of our proposed method on real-world datasets through a comparative study with benchmark techniques. The study consists of two parts, comparison with monotonic neural networks and comparison with methods incorporating both monotonicity and convexity. For the first part, five datasets were used. The Auto-MPG(UCI Machine Learning Repository, 1983–2021) and Blog Feedback (Spiliopoulou et al., 2014) datasets were used for regression tasks, while the Heart Disease (UCI Machine Learning Repository, 1988–2021), COMPAS (Angwin et al., 2016), and Loan Defaulter (Wendy Kan / Kaggle, 2024) datasets were used for classification tasks. The benchmark methods in this comparison include the most recent monotonic neural networks approaches reviewed in Section 2. For the second part, the Car Sales (hsinha53 / Kaggle, 2023), Puzzle Sales (dbahri / Kaggle, 2024a), and Wine Quality (dbahri / Kaggle, 2024b) datasets were used, all of which were designed for regression tasks. The benchmark methods selected for comparison were SCNN and PenDer, as they incorporate both convexity and monotonic-convexity constraints. For regression tasks, we reported metrics including mean squared error (MSEs) and root mean squared error (RMSE), while for classification tasks, we reported accuracy. Further details

Table 2: Results on real-world datasets for comparison with monotonic neural networks

| Method | Auto MPG | Heart Disease | COMPAS | Blog Feedback | Loan Defaulter |
|---|---|---|---|---|---|
| | Test MSE ↓ | Test Acc ↑ | Test Acc ↑ | Test RMSE ↓ | Test Acc ↑ |
| DLN You et al. (2017) | $13.34 \pm 2.42$ | $0.86 \pm 0.02$ | $67.9 \pm 0.3$ | $0.161 \pm 0.001$ | $65.1 \pm 0.2$ |
| Min-Max Net Daniels & Velikova (2010) | $10.14 \pm 1.54$ | $0.75 \pm 0.04$ | $67.8 \pm 0.1$ | $0.163 \pm 0.001$ | $64.9 \pm 0.1$ |
| Non-Neg-DNN | $- - -$ | $- - -$ | $67.3 \pm 0.9$ | $0.168 \pm 0.001$ | $65.1 \pm 0.1$ |
| COMET Sivaraman et al. (2020) | $8.81 \pm 1.81$ | $0.86 \pm 0.03$ | $- - -$ | $- - -$ | $- - -$ |
| Certified MNN Liu et al. (2020) | $- - -$ | $- - -$ | $68.8 \pm 0.9$† | $0.158 \pm 0.001$ | $65.2 \pm 0.1$ |
| LMN Nolte et al. (2022) | $7.58 \pm 1.20$† | $\mathbf{0.90 \pm 0.02}$ | $69.3 \pm 0.1$† | $0.160 \pm 0.001$ | $\mathbf{65.4 \pm 0.0}$ |
| Constrained MNN Runje & Shankaranarayana (2023) | $8.37 \pm 0.08$ | $0.89 \pm 0.00$† | $69.2 \pm 0.2$† | $0.156 \pm 0.001$ | $65.3 \pm 0.1$† |
| SMNN Kim & Lee (2024) | $7.44 \pm 1.20$† | $0.88 \pm 0.04$† | $69.3 \pm 0.9$† | $\mathbf{0.150 \pm 0.001}$ | $65.0 \pm 0.1$ |
| **COMONet (Ours)** | $\mathbf{7.38 \pm 1.32}$ | $0.87 \pm 0.04$† | $\mathbf{69.5 \pm 1.0}$ | $0.153 \pm 0.001$ | $64.9 \pm 0.1$ |

Table 3: Results on real-world datasets for comparison with SCNN and PenDer

| Method | Car Sales (Test MSE ↓) | | Puzzle Sales (Test MSE ↓) | | Wine Quality (Test MSE ↓) | |
|---|---|---|---|---|---|---|
| | (conv) | (conv, decr) | (conc) | (conc, incr) | (conc) | (conc, incr) |
| SCNN Gupta et al. (2018) | $11093 \pm 487$ | $10880 \pm 291$ | $9460 \pm 256$† | $\mathbf{9258 \pm 319}$ | $6.32 \pm 0.19$ | $6.43 \pm 0.18$ |
| PenDer Gupta et al. (2021) | $10411 \pm 107$† | $10415 \pm 104$† | $9428 \pm 113$† | $9519 \pm 92$† | $\mathbf{5.19 \pm 0.11}$ | $5.27 \pm 0.20$† |
| **COMONet (Ours)** | $\mathbf{10391 \pm 140}$ | $\mathbf{10410 \pm 128}$ | $\mathbf{9409 \pm 41}$ | $9263 \pm 86$† | $5.53 \pm 0.46$† | $\mathbf{5.26 \pm 0.06}$ |

on the experiments and additional information about the datasets can be found in Appendix E. All experiments in this section were conducted over multiple iterations, with the mean and standard deviation reported. The best performance for each dataset is highlighted in bold, and dagger symbol (†) indicates statistical tie with the best-performing method. We consider two methods to be in a statistical tie when their mean test MSE ± one standard deviation intervals overlap.

**Comparison with monotonic neural networks:** The results shown in Table 2 represent the means and standard deviations obtained from cross-validation. As shown in Table 2, our method generally performed well, achieving the best performance on some datasets and remaining comparable to other methods on the rest. Specifically, it achieved the best results for the Auto-MPG and COMPAS datasets. For the Heart Disease dataset, it was statistically tied with the best-performing method. On the Blog Feedback dataset, it ranked second. Although its ranking for the Loan Defaulter dataset was lower, its accuracy remained within a reasonable range compared to other methods.

**Comparison with SCNN and PenDer:** Our experimental evaluation compares COMONet against SCNN and PenDer on three real-world datasets—Car Sales, Puzzle Sales, and Wine Quality—under two constraint settings per dataset: convex (concave) only, and convex (concave) monotonic. For each dataset, we used the provided train/test split and averaged the test MSE over five independent runs using the optimal hyperparameter settings found. Table 3 shows performance of the proposed method and the comparison methods on these three datasets. COMONet achieves the best Test MSE in four of the six settings Car Sales (conv), (conv, decr), Puzzle Sales (conc) and Wine Quality (conc, incr), and when accounting for statistical ties matches or outperforms all baselines across all six settings. While PenDer matches or outperforms across all datasets and settings, its shape-conformance metrics $\mathcal{M}_k$ and $\mathcal{C}_k$ sometimes fall below 1, indicating that it fails to fully satisfy the prescribed constraints. Here, $\mathcal{M}_k$ and $\mathcal{C}_k$ denote the proportions of samples satisfying monotonicity and convexity constraints respectively (Gupta et al., 2021). For example, the convexity score on the Puzzle Sales dataset and both the monotonicity and convexity scores on the Wine Quality dataset are 0.98 or 0.99—values close to one but nevertheless indicative of incomplete constraint satisfaction. Table 8 in Appendix F.2 shows the detailed numerical results for PenDer's performance and its constraint conformance. These results demonstrate that COMONet not only delivers comparable or superior predictive performance but also guarantees full adherence to the enforced shape constraints.

## 6 CONCLUSION

In this work, we introduced COMONet, a gradient-descent–trained neural architecture that embeds domain knowledge as inductive biases—enforcing convexity, concavity, monotonicity, and their combinations—while permitting selective application of these shape constraints per variable. empirical results on synthetic and real-world datasets demonstrate that COMONet not only matches or exceeds the predictive performance of existing baselines but also guarantees strict adherence to the specified constraints. However, COMONet requires a priori knowledge of each variable's shape constraints, meaning incomplete or erroneous domain information may impair its effectiveness. In addition, while COMONet focuses on global variable-wise shape constraints, extending the framework toward conditional or interaction-dependent shape behaviors represents a technically challenging yet promising direction. Looking ahead, we will seek theoretical guarantees that COMONet serves as a universal approximator for arbitrary functions under prescribed shape constraints and will explore its use as a modular component in time-series and image-based tasks.

## 7 REPRODUCIBILITY STATEMENT

To ensure reproducibility of our work, we provide the full implementation of our experiments as supplementary material, enabling others to directly verify and replicate our results. Detailed descriptions of the experimental settings and procedures are included in Appendix E, covering datasets, preprocessing step, model configurations, and training protocols. In addition, all theoretical proofs supporting our methods are presented in Appendix A for completeness and clarity. Together, these resources are intended to facilitate transparent and reproducible validation of our findings.

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

# A    PROOFS

**Lemma 4.1**   Let $h_{\mathrm{conv}} : \mathbb{R}^d \to \mathbb{R}^k$ and denote its $j$th coordinate by $f_j(\mathbf{t}) := \big[h_{\mathrm{conv}}(\mathbf{t})\big]_j = \big(\exp(\mathbf{w}_j)^\top \mathbf{t} + b_j\big)_+$. Then, $\forall j \in [k]$, $f_j$ is jointly convex in $\mathbf{t}$ and $f_j$ is coordinatewise increasing, i.e. $\frac{\partial f_j}{\partial t_i} \geq 0$, $\forall i \in [d]$.

*Proof.* Write the vector-valued map as $f_j(\mathbf{t}) = \sigma\big(g(\mathbf{t})\big)$ with

$$g(\mathbf{t}) = \exp(\mathbf{w}_j)^\top \mathbf{t} + b_j \ \in \ \mathbb{R}^k, \qquad \sigma(\mathbf{z}) = (\mathbf{z})_+ \ = \ \max(0, \mathbf{z}) \quad \text{(applied element-wise).}$$

Where $\exp(\mathbf{w}_j)^\top$ and $b_j$ indicates $j$th column vector of weight matrix $\exp(\mathbf{W})$ and $j$th elements of bias vector $\mathbf{b}$. $g$ is an affine map, hence jointly convex in $\mathbf{t}$ (Boyd & Vandenberghe, 2004; Rockafellar, 1997). And $\exp(\mathbf{W})$ has strictly positive weights, so

$$\nabla g_j(\mathbf{t}) = \exp(\mathbf{w}_j) \geq 0. \,\text{(element-wise)}$$

And $\sigma : \mathbb{R}^k \to \mathbb{R}^k$ is convex and increasing coordinatewise (since $\sigma'(z) \in \{0, 1\} \geq 0$). By the standard result that composition of a convex affine map and a convex increasing scalar function is jointly convex(Rockafellar, 1997). $f_j(\mathbf{t}) := \big[h_{\mathrm{conv}}(\mathbf{t})\big]_j = \big(\exp(\mathbf{w}_j)^\top \mathbf{t} + b_j\big)_+$ is convex. Furthermore, by the chain rule,

$$\frac{\partial f_j}{\partial x_i} = \sigma'\big(g(\mathbf{t})\big) \cdot \exp(w_{j,i}) \ \geq \ 0,$$

where $\exp(w_{j,i})$ refers $i$th elements of $\exp(\mathbf{w}_j)$, showing coordinate-wise monotonicity. So, $f_j(\mathbf{t})$ is jointly convex and coordinate-wise monotonically increasing in $\mathbf{t}$. □

**Lemma 4.2**   Let $h_{\mathrm{relu}} : \mathbb{R}^d \to \mathbb{R}^k$ and denote its $j$th coordinate by $f_j(\mathbf{t}) := \big[h_{\mathrm{relu}}(\mathbf{t})\big]_j = \big(\mathbf{w}_j^\top \mathbf{t} + b_j\big)_+$. Then, $\forall j \in [k]$, $f_j$ is jointly convex in $\mathbf{t}$.

*Proof.* Write the vector-valued map as $f_j(\mathbf{t}) = \sigma\big(g(\mathbf{t})\big)$ with

$$g(\mathbf{t}) = \mathbf{w}_j^\top \mathbf{t} + b_j \ \in \ \mathbb{R}^k, \qquad \sigma(\mathbf{z}) = (\mathbf{z})_+ \ = \ \max(0, \mathbf{z}) \quad \text{(applied element-wise).}$$

Where $\mathbf{w}_j^\top$ and $b_j$ indicates $j$th column vector of weight matrix $\exp(\mathbf{W})$ and $j$th elements of bias vector $\mathbf{b}$. $g$ is affine in $\mathbf{t}$, hence jointly convex. And $\sigma : \mathbb{R}^k \to \mathbb{R}^k$ is convex and coordinate-wise increasing since, $\sigma'(z) \in \{0, 1\} \geq 0$. By the standard result that composition of a convex affine map and a convex increasing scalar function is jointly convex. By standard results on composition, $f_j(\mathbf{t})$ is jointly convex in $\mathbf{t}$. □

**Theorem 4.6.** Let $f(\mathbf{x})$ be the proposed COMONet, which has $l$ hidden layers. Partition the input $\mathbf{x} \in \mathbb{R}^d$ as $\mathbf{x} = (\mathbf{x}_v, \mathbf{x}_{\neg v})$, $\mathbf{x}_v = \{x_i \mid i \in \mathcal{V}\}$, $\mathcal{V} \subseteq [d]$. Then $f(\mathbf{x})$ is *partially jointly convex* with respect to $\mathbf{x}_v$.

*Proof.* Let $\mathbf{x} = (\mathbf{x}_v, \mathbf{x}_{\neg v})$, $\mathbf{x}_v = (\mathbf{x}_{cv}, \mathbf{x}_{mv})$ so that equivalently $\mathbf{x} = (\mathbf{x}_{cv}, \mathbf{x}_{mv}, \mathbf{x}_{\neg v})$. In this composition, convex features $\mathbf{x}_{cv}$ feeds into a $h_{\mathrm{relu}}$-then-$h_{\mathrm{conv}}$ chain, where as convex-monotonic features $\mathbf{x}_{mv}$ feeds into a $h_{\mathrm{conv}}$ chain. We show that every layer is jointly convex in $\mathbf{x}_v$, which implies that the entire network is jointly convex in $\mathbf{x}_v$. There are two cases:

Case 1: convex features, $\mathbf{x}_{cv}$

First layer on $\mathbf{x}_{cv}$ is $h_{\mathrm{relu}}$:

$$\mathbf{z}^{(1)} = h_{\mathrm{relu}}^{(1)}(\mathbf{x}_{cv}),$$

which by lemma 4.2 is jointly convex in $\mathbf{x}_{cv}$. Subsequent layers along any path to the output are convex-units:

$$\mathbf{z}^{(i)} = h_{\mathrm{conv}}^{(i)}(\mathbf{z}^{(i-1)}, \ \dots),$$

which by lemma 4.1 is also jointly convex and increase in $x_i$. Composition of a convex map and an affine/increasing convex map remains convex. Hence any path from $\mathbf{x}_{cv}$ through $h_{\text{relu}}$-then-$h_{\text{conv}}$ units which follows the fully connected layer (affine transform) :

$$f = \exp(\mathbf{w}_j)^\top \mathbf{z}^{(l)} + b_j$$

is jointly convex in $\mathbf{x}_{cv}$.

Case 2: convex-monotonic features, $\mathbf{x}_{mv}$.

First layer on $\mathbf{x}_{mv}$ is $h_{\text{conv}}$:

$$\mathbf{z}^{(1)} = h_{\text{conv}}^{(1)}(\mathbf{x}_{mv}),$$

which by lemma 4.2 is jointly convex in $\mathbf{x}_{mv}$. Subsequent layers along any path to the output are also $h_{\text{conv}}$:

$$\mathbf{z}^{(i)} = h_{\text{conv}}^{(i)}(\mathbf{x}_{mv}),$$

Composition of a convex map and an affine/increasing convex map remains convex. Hence any path from $x_i$ through $h_{\text{conv}}$ which follows the fully connected layer (affine transform) :

$$f = \exp(\mathbf{w}_j)^\top \mathbf{z}^{(l)} + b_j$$

is jointly convex in $\mathbf{x}_{mv}$.

Finally, note that every layer of COMONet receives the concatenated block $\mathbf{x}_v = (\mathbf{x}_{cv}, \mathbf{x}_{mv})$ only through an affine maps. Since affine maps preserve joint convexity and all subsequent activations are convex and coordinatewise nondecreasing, the layerwise composition remains jointly convex in the entire block $\mathbf{x}_v$. Hence, $f$ is jointly convex in $\mathbf{x}_v$. $\qquad\square$

**Lemma 4.3** Let $h_{\text{conc}} : \mathbb{R}^d \to \mathbb{R}^k$ and denote its $j$th coordinate by $f_j(\mathbf{t}) := \big[h_{\text{conv}}(\mathbf{t})\big]_j = -\big(-\exp(\mathbf{w}_j)^\top \mathbf{t} - b_j\big)_+$. Then, $\forall j \in [k]$, $f_j$ is jointly concave in $\mathbf{t}$ and $f_j$ is coordinate-wise increasing, i.e. $\frac{\partial f_j}{\partial t_i} \geq 0, \forall i \in [d]$.

*Proof.* Write the vector-valued map as $f_j(\mathbf{t}) = \sigma\big(g(\mathbf{t})\big)$ with

$$g(\mathbf{t}) = -\exp(\mathbf{w}_j)^\top \mathbf{t} - b_j \in \mathbb{R}^k, \qquad \sigma(\mathbf{z}) = -(\mathbf{z})_+ = -\max(0, \mathbf{z}) \quad \text{(applied element-wise).}$$

Where $\exp(\mathbf{w}_j)^\top$ and $b_j$ indicates $j$th column vector of weight matrix $\exp(\mathbf{W})$ and $j$th elements of bias vector $\mathbf{b}$. $g$ is an affine map, hence jointly concave in $\mathbf{t}$ (Boyd & Vandenberghe, 2004; Rockafellar, 1997). And $\exp(\mathbf{W})$ has strictly negative weights, so

$$\nabla g_j(\mathbf{t}) = -\exp(\mathbf{w}_j) \leq 0. \,\text{(element-wise)}$$

And $\sigma : \mathbb{R}^k \to \mathbb{R}^k$ is concave and decreasing coordinate-wise (since $\sigma'(z) \in \{-1, 0\} \leq 0$). By the standard result that composition of a concave affine map and a concave decreasing scalar function is jointly concave. $f_j(\mathbf{t}) := \big[h_{\text{conc}}(\mathbf{t})\big]_j = -\big(-\exp(\mathbf{w}_j)^\top \mathbf{t} - b_j\big)_+$ is concave. Furthermore, by the chain rule,

$$\frac{\partial f_j}{\partial x_i} = \sigma'\big(g(\mathbf{t})\big) \cdot \exp(w_{j,i}) \geq 0,$$

where $\exp(w_{j,i})$ refers $i$th elements of $\exp(\mathbf{w}_j)$, showing coordinate-wise monotonicity. So, $f_j(\mathbf{t})$ is jointly concave and coordinate-wise monotonically increasing in $\mathbf{t}$. $\qquad\square$

**Lemma 4.4** Let $h_{\text{ref-relu}} : \mathbb{R}^d \to \mathbb{R}^k$ and denote its $j$th coordinate by $f_j(\mathbf{t}) := \big[h_{\text{relu}}(\mathbf{t})\big]_j = -\big(-\mathbf{w}_j^\top \mathbf{t} - b_j\big)_+$. Then, $\forall j \in [k]$, $f_j$ is jointly concave in $\mathbf{t}$.

*Proof.* Write the vector-valued map as $f_j(\mathbf{t}) = \sigma\big(g(\mathbf{t})\big)$ with

$$g(\mathbf{t}) = -\mathbf{w}_j^\top \mathbf{t} - b_j \in \mathbb{R}^k, \qquad \sigma(\mathbf{z}) = -(\mathbf{z})_+ = -\max(0, \mathbf{z}) \quad \text{(applied element-wise).}$$

Where $\mathbf{w}_j^\top$ and $b_j$ indicates $j$th column vector of weight matrix $\exp(\mathbf{W})$ and $j$th elements of bias vector $\mathbf{b}$. $g$ is affine in $\mathbf{t}$, hence jointly concave. And $\sigma : \mathbb{R}^k \to \mathbb{R}^k$ is concave and coordinate-wise decreasing since, $\sigma'(z) \in \{-1, 0\} \leq 0$. By the standard result that composition of a concave affine map and a concave decreasing scalar function is jointly concave. By standard results on composition, $f_j(\mathbf{t})$ is jointly concave in $\mathbf{t}$. $\qquad\square$

**Theorem 4.7.** Let $f(\mathbf{x})$ be the proposed COMONet, which has $l$ hidden layers. Partition the input $\mathbf{x} \in \mathbb{R}^d$ as $\mathbf{x} = (\mathbf{x}_c, \mathbf{x}_{\neg c})$, $\mathbf{x}_c = \{x_i \mid i \in \mathcal{C}\}$, $\mathcal{C} \subseteq [d]$. Then $f(\mathbf{x})$ is *partially jointly concave* with respect to $\mathbf{x}_c$.

*Proof.* Let $\mathbf{x} = (\mathbf{x}_c, \mathbf{x}_{\neg c})$, $\mathbf{x}_c = (\mathbf{x}_{cc}, \mathbf{x}_{mc})$ so that equivalently $\mathbf{x} = (\mathbf{x}_{cc}, \mathbf{x}_{mc}, \mathbf{x}_{\neg c})$. In this composition, concave features $\mathbf{x}_{cc}$ feeds into a $h_{\text{ref-relu}}$-then-$h_{\text{conc}}$ chain, where as concave-monotonic features $\mathbf{x}_{mc}$ feeds into a $h_{\text{conc}}$ chain. We show that every layer is jointly concave in $\mathbf{x}_c$, which implies that the entire network is jointly concave in $\mathbf{x}_c$. There are two cases:

Case 1: concave features, $\mathbf{x}_{cc}$

First layer on $\mathbf{x}_{cc}$ is $h_{\text{ref-relu}}$:
$$\mathbf{z}^{(1)} = h_{\text{ref-relu}}^{(1)}(\mathbf{x}_{cc}),$$
which by lemma 4.4 is jointly concave in $\mathbf{x}_{cc}$. Subsequent layers along any path to the output are concave-units:
$$\mathbf{z}^{(i)} = h_{\text{conc}}^{(i)}(\mathbf{z}^{(i-1)}, \dots),$$
which by lemma 4.3 is also jointly concave and increase in $\mathbf{x}_{cc}$. Composition of a concave map and an affine/increasing concave map remains concave. Hence any path from $x_i$ through $h_{\text{ref-relu}}$-then-$h_{\text{conc}}$ units which follows the fully connected layer (affine transform) :
$$f = \exp(\mathbf{w}_j)^\top \mathbf{z}^{(l)} + b_j$$
is jointly concave in $\mathbf{x}_{cc}$.

Case 2: concave-monotonic features, $\mathbf{x}_{mc}$

First layer on $\mathbf{x}_{mc}$ is $h_{\text{conc}}$:
$$\mathbf{z}^{(1)} = h_{\text{conc}}^{(1)}(x_i, \dots),$$
which by lemma 4.4 is jointly concave in $\mathbf{x}_{mc}$. Subsequent layers along any path to the output are also $h_{\text{conc}}$:
$$\mathbf{z}^{(i)} = h_{\text{conc}}^{(i)}(\mathbf{z}^{(i-1)}, \dots),$$
Composition of a concave map and an affine/increasing concave map remains concave. Hence any path from $\mathbf{x}_{mc}$ through $h_{\text{conc}}$ which follows the fully connected layer (affine transform) :
$$f = \exp(\mathbf{w}_j)^\top \mathbf{z}^{(l)} + b_j$$
is jointly concave in $\mathbf{x}_{mc}$.

Finally, note that every layer of COMONet receives the concatenated block $\mathbf{x}_c = (\mathbf{x}_{cc}, \mathbf{x}_{mc})$ only through an affine maps. Since affine maps preserve joint concavity and all subsequent activations are concave and coordinatewise nondecreasing, the layerwise composition remains jointly concave in the entire block $\mathbf{x}_c$. Hence, $f$ is jointly concave in $\mathbf{x}_c$. □

**Lemma 4.5** Let $h_{\text{mono}} : \mathbb{R}^d \to \mathbb{R}^k$ and denote its $j$th coordinate by $f_j(\mathbf{t}) := \big[h_{\text{mono}}(\mathbf{t})\big]_j = \big(\exp(\mathbf{w}_j)^\top \mathbf{t} + b_j\big)_+^n$. Then, $\forall j \in [k]$, $f_j$ is coordinate-wise increasing in $\mathbf{t}$, i.e. $\frac{\partial f_j}{\partial t_i} \geq 0$, $\forall i \in [d]$.

*Proof.* Write the vector-valued map as $f_j(\mathbf{t}) = \sigma\big(g(\mathbf{t})\big)$ with

$g(\mathbf{t}) = \exp(\mathbf{w}_j)^\top \mathbf{t} + b_j \in \mathbb{R}^k$, $\sigma(\mathbf{z}) = (\mathbf{z})_+^n = \min(\max(0, \mathbf{z}), n)$ (applied element-wise).

Where $\exp(\mathbf{w}_j)^\top$ and $b_j$ indicates $j$th column vector of weight matrix $\exp(\mathbf{W})$ and $j$th elements of bias vector $\mathbf{b}$. $g$ is an affine map, $\exp(\mathbf{W})$ has strictly positive weights, so
$$\nabla g_j(\mathbf{t}) = \exp(\mathbf{w}_j) \geq 0. \text{ (element-wise)}$$
And $\sigma : \mathbb{R}^k \to \mathbb{R}^k$ is monotonically increasing coordinate-wise (since $\sigma'(z) \in \{0, 1\} \geq 0$). Furthermore, by the chain rule,
$$\frac{\partial f_j}{\partial x_i} = \sigma'\big(g(\mathbf{t})\big) \cdot \exp(w_{j,i}) \geq 0,$$
where $\exp(w_{j,i})$ refers $i$th elements of $\exp(\mathbf{w}_j)$, showing coordinate-wise monotonicity. So, $f_j(\mathbf{t})$ is coordinate-wise monotonic increasing in $\mathbf{t}$. □

**Theorem 4.8.** Let $f(\mathbf{x})$ be the proposed COMONet, which has $l$ hidden layers. Partition the input $\mathbf{x} \in \mathbb{R}^d$ as $\mathbf{x} = (\mathbf{x}_m, \mathbf{x}_{\neg m})$, $\mathbf{x}_m = \{x_i \mid i \in \mathcal{M}\}$, $\mathcal{M} \subseteq [d]$. Then $f(\mathbf{x})$ is *partially monotonic increasing* with respect to $\mathbf{x}_m$. In particular, for each $x_i$ with $i \in \mathcal{M}$, $f$ is monotonic (increasing) in $x_i$.

*Proof.* Let $\mathbf{x} = (\mathbf{x}_m, \mathbf{x}_{\neg m})$, $\mathbf{x}_m = (\mathbf{x}_{mn}, \mathbf{x}_{mv}, \mathbf{x}_{mc})$ so that equivalently $\mathbf{x} = (\mathbf{x}_{mn}, \mathbf{x}_{mv}, \mathbf{x}_{mc}, \mathbf{x}_{\neg m})$. We will show that for each fixed setting of all coordinates except a single $x_i$ with $i \in M$, the scalar output $f(\mathbf{x})$ is monotonically increasing in $x_i$. There are three cases:

Case 1: monotonic features $x_i \in \mathbf{x}_{mn}$

First layer on $x_i$ is $h_{\text{mono}}$:

$$\mathbf{z}^{(1)} = h_{\text{mono}}^{(1)}(x_i, \dots),$$

which by lemma 4.5 is monotonically increasing in $x_i$, Subsequent hidden layers along any path to the output are consist by monotonic units ($h_{\text{mono}}$), convex units ($h_{\text{conv}}$) and concave units ($h_{\text{conc}}$) for $k = 2, ..., l$ :

$$\mathbf{z}^{(k)} = h_{\text{mono}}^{(k)}\big(\mathbf{z}^{(k-1)}, \dots\big) \quad \text{or} \quad \mathbf{z}^{(k)} = h_{\text{conv}}^{(k)}\big(\mathbf{z}^{(k-1)}, \dots\big) \quad \text{or} \quad \mathbf{z}^{(k)} = h_{\text{conc}}^{(k)}\big(\mathbf{z}^{(k-1)}, \dots\big).$$

By lemma 4.5, lemma 4.1 and lemma 4.3, each of these three unit types has nonnegative partial derivatives in all its inputs. Hence at every hidden layer $k$, along every path, we have

$$\frac{\partial h_j^{(k)}}{\partial x_i} \geq 0.$$

The layer-wise computation thus proceeds up to the final hidden layer, indexed $k = l$. There, the network produces the feature vector $\mathbf{z}^{(l)}$, which is then passed through the output affine map with strictly positive weights:

$$f = \exp(\mathbf{w}_j)^\top \mathbf{z}^{(l)} + b_j$$

Because each entry of $\exp(\mathbf{w}_j)^\top$ is positive, the total derivative is a positive weighted sum of nonnegative terms. Therefore

$$\frac{\partial f}{\partial x_i} \geq 0,$$

showing that output of $f$ is monotonically increasing in $x_i$ for every $x_i \in \mathbf{x}_{mn}$.

Case 2: convex and monotonic features, $x_i \in \mathbf{x}_{mv}$

Every layer along its path is either a convex unit $h_{\text{conv}}$ or the fully connected layer. By lemma 4.1, each $h_{\text{conv}}$ has nonnegative partial derivatives, and the fully connected layer does as well. Hence $f$ is monotonically increasing in $x_i$ for every $x_i \in \mathbf{x}_{mv}$.

Case 3: concave and monotonic features, $x_i \in \mathbf{x}_{mc}$

Every layer along its path is either a concave unit $h_{\text{conc}}$ or the fully connected layer. By lemma 4.3, each $h_{\text{conc}}$ has nonnegative partial derivatives, and the fully connected layer does as well. Hence $f$ is monotonically increasing in $x_i$ for every $x_i \in \mathbf{x}_{mc}$.

Finally, in all cases, $\frac{\partial f}{\partial x_i} \geq 0$ for every $x_i \in \mathbf{x}_m$. Hence $f$ is partially monotonic increasing in $\mathbf{x}_m$. $\square$

**Theorem 4.9.** Let $f : \mathbb{R}^2 \to \mathbb{R}$ be twice differentiable and satisfy: for all $x_2 \in \mathbb{R}$, the mapping $x_1 \mapsto f(x_1, x_2)$ is convex, and for all $x_1 \in \mathbb{R}$, the mapping $x_2 \mapsto f(x_1, x_2)$ is concave. Then, if we decompose

$$f(x_1, x_2) = g(x_1) + h(x_2) + \phi(x_1, x_2),$$

where $g$ depends only on $x_1$ and $h$ depends only on $x_2$, then the pure interaction term $\phi$, which preserves the convex–concave assignments for any admissible choices of $g$ and $h$, must be

$$\phi(x_1, x_2) = \alpha\, x_1 x_2, \qquad \alpha \in \mathbb{R}.$$

*Proof.* Since $f$ is twice differentiable, consider its Hessian:

$$H_f(x_1, x_2) = \begin{pmatrix} f_{11} & f_{12} \\ f_{21} & f_{22} \end{pmatrix}, \quad f_{ij} = \frac{\partial^2 f}{\partial x_i \partial x_j}.$$

Convexity in $x_1$ implies $f_{11}(x_1, x_2) \geq 0$ and concavity in $x_2$ implies

$$f(x_1, x_2) = g(x_1) + h(x_2) + \phi(x_1, x_2),$$

where $g$ and $h$ collect all single-variable terms and $\phi$ denotes the pure interaction. Since the magnitudes of $g''(x_1) \geq 0$ and $h''(x_2) \leq 0$ are arbitrary (learned from data), preserving the convex–concave curvature assignments for *all* admissible choices of $g''$ and $h''$ requires that $\phi$ contribute no curvature:

$$\phi_{11}(x_1, x_2) = 0, \qquad \phi_{22}(x_1, x_2) = 0.$$

Integrating $\phi_{11} = 0$ twice with respect to $x_1$ gives

$$\phi(x_1, x_2) = A(x_2)\, x_1 + B(x_2).$$

Differentiating twice with respect to $x_2$ and using $\phi_{22} = 0$ yields

$$A''(x_2)x_1 + B''(x_2) = 0,$$

so $A''(x_2) = 0$ and $B''(x_2) = 0$. Hence

$$A(x_2) = a_1 x_2 + a_0, \qquad B(x_2) = b_1 x_2 + b_0.$$

Absorbing all single-variable terms into $g$ and $h$ leaves only

$$\phi(x_1, x_2) = a_1 x_1 x_2.$$

Thus the interaction term must be of the bilinear form $\alpha x_1 x_2$, completing the proof. $\qquad\square$

**Theorem 4.10.** Let $f : \mathbb{R}^2 \to \mathbb{R}$ be twice differentiable and jointly convex (or jointly concave) in $(x_1, x_2)$, and assume that $x_1$ and $x_2$ belong to the same curvature group of COMONet (both $\mathcal{CV}$ or both $\mathcal{CC}$). Consider adding the bilinear term

$$\phi(x_1, x_2) = \beta\, x_1 x_2, \qquad \beta \in \mathbb{R},$$

and redefine the mapping by

$$f(x_1, x_2) := f(x_1, x_2) + \phi(x_1, x_2).$$

Then this addition preserves the assigned per-variable constraints while allowing the joint convexity (or concavity) in $(x_1, x_2)$ to be relaxed.

*Proof.* Consider the added interaction $\phi(x_1, x_2) = \beta x_1 x_2$. Its second-order partial derivatives are

$$\phi_{11}(x_1, x_2) = 0, \qquad \phi_{22}(x_1, x_2) = 0, \qquad \phi_{12}(x_1, x_2) = \phi_{21}(x_1, x_2) = \beta.$$

Thus, adding $\phi$ does not modify the curvature of $f$ with respect to each individual variable, since the per-variable second derivatives remain unchanged:

$$(f + \phi)_{11} = f_{11}, \qquad (f + \phi)_{22} = f_{22}.$$

However, the mixed derivative becomes

$$(f + \phi)_{12} = f_{12} + \beta,$$

so the joint curvature in $(x_1, x_2)$ is relaxed. Therefore, adding the bilinear term preserves the assigned per-variable curvature signs and allows the joint curvature to change. $\qquad\square$

# B    FLOW DIAGRAMS OF COMONET

Fig. 7 present the flow structure of COMONet to demonstrate how each variable group ($\mathbf{x}_{cv}$, $\mathbf{x}_{mv}$, $\mathbf{x}_{cc}$, $\mathbf{x}_{mc}$, $\mathbf{x}_{mn}$, $\mathbf{x}_u$) contributes to the final prediction through their respective computational flows. Each subfigure highlights the specific path for a variable group, represented by bold dashed lines, showing how the input is processed through layers to produce the final output. This detailed visualization helps to clarify the role and influence of each group of variables in the model's overall architecture.

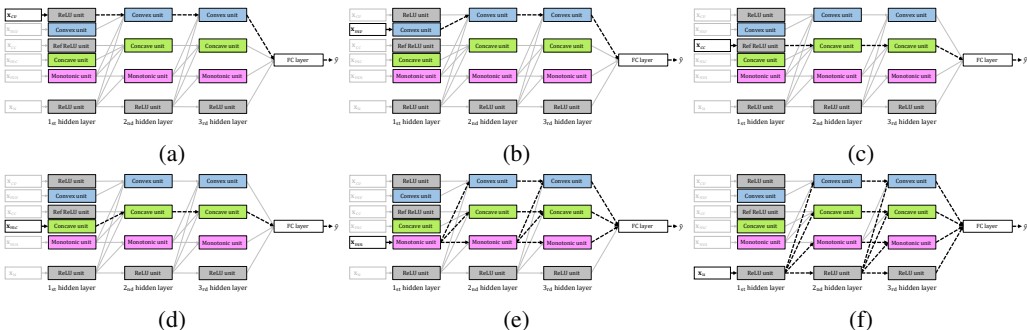

Figure 7: Flow diagrams representing the computational flows for each variable group in COMONet. Bold dashed lines indicate the paths followed by individual variable groups (a) $\mathbf{x}_{cv}$, (b) $\mathbf{x}_{mv}$, (c) $\mathbf{x}_{cc}$, (d) $\mathbf{x}_{cv}$, (e) $\mathbf{x}_{mn}$, (f) $\mathbf{x}_u$.

# C    INTERACTION LAYER OF COMONET

Fig. 8 and shows the all needed pairwise cross interactions between $\mathbf{x}_{cv}$ and $\mathbf{x}_{cc}$. And, Fig. 9 and shows the all possible pairwise intra interactions between $\mathbf{x}_{cc}$-$\mathbf{x}_{cc}$ or $\mathbf{x}_{cv}$-$\mathbf{x}_{cv}$.

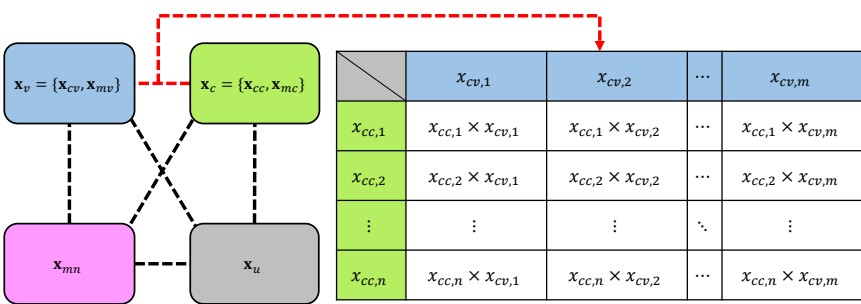

Figure 8: Cross interaction terms between convex and concave features could be captured by Interaction layer.

Figure 9: Intra-interaction terms between Convex-convex and Concave-concave features could be captured by Interaction layer.

# D  CONFIGURATION FLEXIBILITY OF COMONET

Fig. 10 and Fig. 11 shows various COMONet configurations, demonstrating the flexibility of the proposed method. Fig. 10 illustrates the $2^4 - 1 = 15$ possible configurations obtained by dividing the variable groups into four categories: $\mathbf{x}_v$, $\mathbf{x}_c$, $\mathbf{x}_{mn}$, $\mathbf{x}_u$. While, table in Fig. 11 shows all $2^6 - 1 = 63$ possible configurations of COMONet. These configurations highlight the ability of COMONet to handle a wide range of input scenarios while maintaining consistent processing through its computational layers. This adaptability ensures that the model can be tailored to specific tasks by including or excluding certain variable groups as needed.

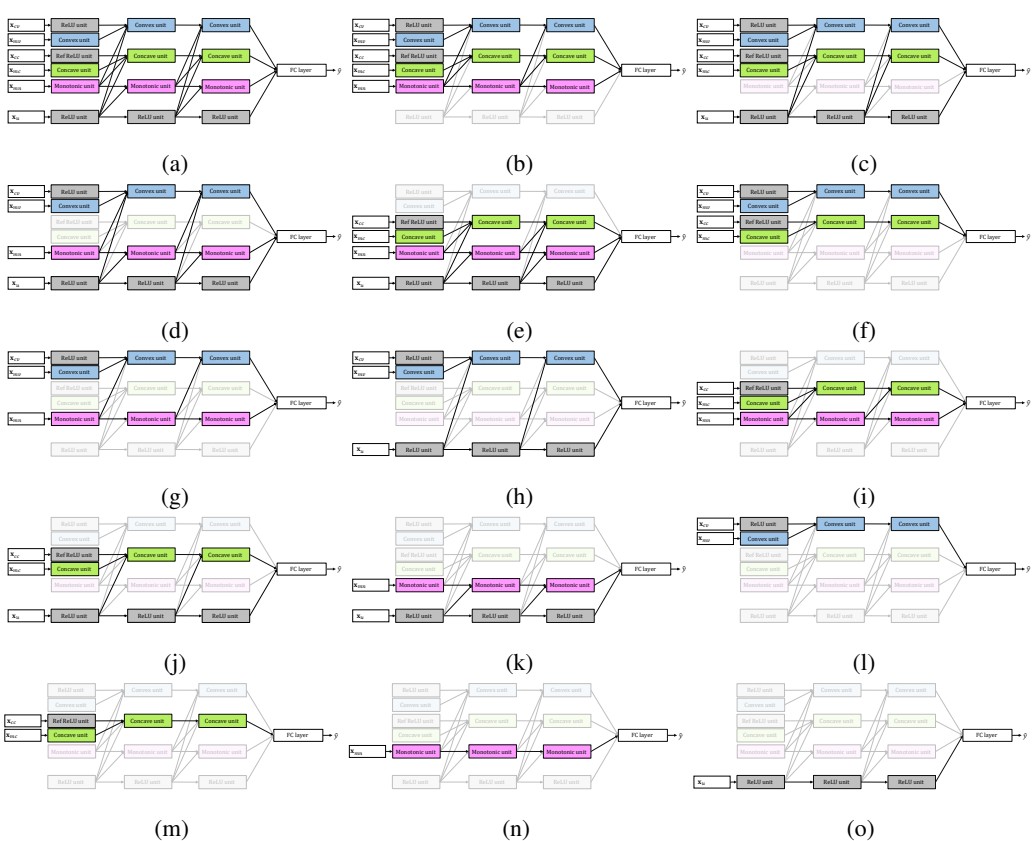

Figure 10: Examples structures of various configurations for COMONet.

| # | $\mathbf{x}_{cv}$ | $\mathbf{x}_{mv}$ | $\mathbf{x}_{cc}$ | $\mathbf{x}_{mc}$ | $\mathbf{x}_{mn}$ | $\mathbf{x}_u$ |
|---|---|---|---|---|---|---|
| | 6 types of features | | | | | |
| 1 | ✓ | ✓ | ✓ | ✓ | ✓ | ✓ |
| | 5 types of features | | | | | |
| 2 | ✓ | ✓ | ✓ | ✓ | ✓ | X |
| 3 | ✓ | ✓ | ✓ | ✓ | X | ✓ |
| 4 | ✓ | ✓ | ✓ | X | ✓ | ✓ |
| 5 | ✓ | ✓ | X | ✓ | ✓ | ✓ |
| 6 | ✓ | X | ✓ | ✓ | ✓ | ✓ |
| 7 | X | ✓ | ✓ | ✓ | ✓ | ✓ |
| | 4 types of features | | | | | |
| 8 | ✓ | ✓ | ✓ | ✓ | X | X |
| 9 | ✓ | ✓ | ✓ | X | ✓ | X |
| 10 | ✓ | ✓ | X | ✓ | ✓ | X |
| 11 | ✓ | X | ✓ | ✓ | ✓ | X |
| 12 | X | ✓ | ✓ | ✓ | ✓ | X |
| 13 | ✓ | ✓ | ✓ | X | X | ✓ |
| 14 | ✓ | ✓ | X | ✓ | X | ✓ |
| 15 | ✓ | X | ✓ | ✓ | X | ✓ |
| 16 | X | ✓ | ✓ | ✓ | X | ✓ |
| 17 | ✓ | ✓ | X | X | ✓ | ✓ |
| 18 | ✓ | X | ✓ | X | ✓ | ✓ |
| 19 | X | ✓ | ✓ | X | ✓ | ✓ |
| 20 | ✓ | X | X | ✓ | ✓ | ✓ |
| 21 | X | ✓ | X | ✓ | ✓ | ✓ |
| 22 | X | X | ✓ | ✓ | ✓ | ✓ |
| | 3 types of features | | | | | |
| 23 | ✓ | ✓ | ✓ | X | X | X |
| 24 | ✓ | ✓ | X | ✓ | X | X |
| 25 | ✓ | X | ✓ | ✓ | X | X |
| 26 | X | ✓ | ✓ | ✓ | X | X |
| 27 | ✓ | ✓ | X | X | ✓ | X |
| 28 | X | ✓ | ✓ | X | ✓ | X |
| 29 | X | ✓ | ✓ | X | X | ✓ |
| 30 | ✓ | X | X | ✓ | ✓ | X |
| 31 | X | ✓ | X | ✓ | ✓ | X |
| 32 | X | X | ✓ | ✓ | ✓ | X |
| 33 | ✓ | ✓ | X | X | X | ✓ |
| 34 | ✓ | X | ✓ | X | X | ✓ |
| 35 | X | ✓ | ✓ | X | X | ✓ |
| 36 | ✓ | X | X | ✓ | X | ✓ |
| 37 | X | ✓ | X | ✓ | X | ✓ |
| 38 | X | X | ✓ | ✓ | X | ✓ |
| 39 | ✓ | X | X | X | ✓ | ✓ |
| 40 | X | ✓ | X | X | ✓ | ✓ |
| 41 | X | X | ✓ | X | ✓ | ✓ |
| 42 | X | X | X | ✓ | ✓ | ✓ |
| | 2 types of features | | | | | |
| 43 | ✓ | X | ✓ | X | X | X |
| 44 | ✓ | X | ✓ | X | X | X |
| 45 | X | ✓ | ✓ | X | X | X |
| 46 | ✓ | X | X | ✓ | X | X |
| 47 | X | ✓ | X | ✓ | X | X |
| 48 | X | X | ✓ | ✓ | X | X |
| 49 | ✓ | X | X | X | ✓ | X |
| 50 | X | ✓ | X | X | ✓ | X |
| 51 | X | X | ✓ | X | ✓ | X |
| 52 | X | X | X | ✓ | ✓ | X |
| 53 | ✓ | X | X | X | X | ✓ |
| 54 | X | ✓ | X | X | X | ✓ |
| 55 | X | X | ✓ | X | X | ✓ |
| 56 | X | X | X | ✓ | X | ✓ |
| 57 | X | X | X | X | ✓ | ✓ |
| | 1 types of features | | | | | |
| 58 | X | X | X | X | ✓ | X |
| 59 | X | X | X | ✓ | X | X |
| 60 | X | X | ✓ | X | X | X |
| 61 | X | X | ✓ | X | X | X |
| 62 | X | ✓ | X | X | X | X |
| 63 | ✓ | X | X | X | X | X |

Figure 11: All possible 63 configurations for COMONet.

# E   DETAILED EXPERIMENT DESCRIPTIONS

## E.1   TRAINING CONFIGURATIONS

All experiments in this study were conducted on a system equipped with an Intel(R) Core(TM) i7-14700K 3.40 GHz processor, 64.0GB of DDR5 RAM, and running Microsoft Windows 11 Pro as the operating system. For GPU computations, we utilized an NVIDIA GeForce RTX 4070 Ti SUPER with 16.0GB of memory. The implementation of all models and experiments was carried out using Python (version 3.12.7) and the PyTorch (Paszke et al., 2019) library (version 2.5.1) with CUDA (version 12.4). During model training, the ADAM (Kingma & Ba, 2014) optimizer was employed as the stochastic optimization solver. Hyperparameters for each dataset were explored using a grid search strategy. For all datasets, the training process incorporated early stopping and, when necessary, exponentiated batch normalization (EBN) to enhance stability and efficiency. Table 4 summarizes the hyperparameter settings for the proposed methods. Hyperparameters for both our proposed and the benchmark methods were selected via grid search over batch sizes of $128, 256, 512$, and learning rates of $0.05, 0.005, 0.002$. The number of epochs varied $[200, 3000]$ depending on the dataset. In general, experiments were conducted using 5-fold cross-validation repeated 5 times, and the mean and standard deviation (std) across 25 runs were reported. However, when the dataset was split into train/validation/test, experiments were conducted 5 times, and the mean and standard deviation from these runs were reported.

Table 4: Hyperparameters of COMONet for real-world datasets

| Dataset | Number of parameters | Learning rate | Batch size |
|---|---|---|---|
| COMPAS | 1457 | 0.005 | 128 |
| Heart Disease | 19649 | 0.002 | 128 |
| Loan Defaulter | 1489 | 0.0005 | 512 |
| Blog Feedback | 5137 | 0.0005 | 256 |
| Auto-MPG | 19265 | 0.005 | 128 |
| Car sales | 1195 | 0.005 | 109 |
| Puzzle sales | 1819 | 0.005 | 155 |
| Wine quality | 6753 | 0.005 | 512 |

Table 5: Hyperparameters of benchmark methods for real-world datasets

| Methods | Dataset | Number of parameters | Learning rate | Batch size |
|---|---|---|---|---|
| SCNN | Car sales | 1450 | 0.005 | 109 |
| SCNN | Puzzle sales | 5460 | 0.005 | 155 |
| SCNN | Wine quality | 9094 | 0.005 | 512 |
| PenDer | Car sales | 6401 | 0.005 | 109 |
| PenDer | Puzzle sales | 6529 | 0.005 | 155 |
| PenDer | Wine quality | 10241 | 0.005 | 512 |

## E.2   STRATEGIES FOR TRAINING STABILIZATION

In this study, one of the key components of the proposed method, the exponentiated weight $(\exp(w))$, has the potential to explode as the weight value increases due to the nature of the exponential function. To address this issue, we adopted the weight initialization strategy introduced in the appendix of SMNN (Kim & Lee, 2024). Specifically, the initial values of the exponentiated weight $w$ were sampled from a uniform distribution within the range $[-20, 2]$, effectively preventing the exploding problem. Furthermore, the scaling parameter $\gamma$ of the Exponentiated Batch Normalization was initialized to 0 to ensure stable training, by making $\exp(\gamma)$ to 1. In addition, for activation functions such as ReLU and ReLU-n, we introduced a Leaky ReLU modification with $\alpha = 0.01$ in their off regions. This adjustment preserves the intended properties of each activation function while

improving training stability.

$$\text{Leaky ReLU-}n(x) = \begin{cases} \alpha x, & \text{if } x < 0, \\ x, & \text{if } 0 \leq x \leq n, \\ \alpha(x - n) + n, & \text{if } x > n. \end{cases} \tag{22}$$

### E.3 EXPONENTIATED BATCH NORMALIZATION

In COMONet, two types of different activation functions (e.g., ReLU and ReLU-$n$) are employed across layers, and the weights in certain layers are exponentiated. These differences in activation types and weight transformations can lead to significant deviations in the value distributions of layer outputs. In some cases, such discrepancies may result in unstable learning dynamics, necessitating the use of Batch Normalization(Ioffe, 2015) to stabilize training.

However, traditional Batch Normalization introduces a scaling parameter, $\gamma$ which can take on negative values during learning ($\gamma < 0$). When $\gamma$ becomes negative, the normalized output may be reversed, violating critical Shape Constraints such as monotonicity or convexity. This sign reversal, in turn, can alter the sign of partial derivatives, fundamentally disrupting structural guarantees for each varlables.

$$y = \frac{x - \mathbb{E}[x]}{\sqrt{Var[x] + \epsilon}} \cdot \exp(\gamma) + \beta \tag{23}$$

To address this issue, we propose Exponentiated Batch Normalization (EBN) equation 23, where the scaling parameter $\gamma$ is replaced with its exponentiated form, $\exp(\gamma)$ when $x$ refers the outputs of the layer and $y$ refers the batch normalized outputs. By enforcing $\exp(\gamma)$ to be strictly positive, we ensure that the normalized output retains its correct sign, thereby preserving the desired Shape Constraints. This approach effectively mitigates the variance in layer output distributions while maintaining stable and consistent training dynamics across heterogeneous layers.

### E.4 DESCRIPTIONS FOR REAL-WORLD DATASETS

This section provides an overview of the real-world datasets used in the experiments. These datasets were derived from previously published benchmarks frequently cited in literature on monotonic and convex neural networks. The criteria for applying shape constraints followed the methodologies outlined in prior benchmark studies. While precisely defining shape constraints poses challenges, as mentioned in the conclusion, future research that focuses on identifying these constraints for specific variables could yield valuable insights. Additionally, some datasets contain instances with relatively small sample sizes, reflecting realistic challenges often encountered in practical applications. Effectively addressing such constraints is critical for developing robust and widely applicable models. Table 6 provides a summarized overview of each dataset, with detailed descriptions presented below (**Bold** text indicates monotonic decrease, while *italic* text denotes concavity.):

**AutoMPG:** The Auto-MPG dataset is a regression dataset with 7 variables and approximately 398 instances, used to predict a car's miles per gallon (mpg). It includes monotonic decreasing relationships between mpg and the variables **weight**, **displacement** and **horse power**.

**Heart Disease:** The Heart Disease dataset is a classification dataset with 13 variables, used to predict the presence or absence of heart disease in individuals. Among the variables, trestbps (resting blood pressure) and chol (cholesterol level) are known to have monotonic increasing relationships with the risk of heart disease.

**COMPAS*:** The COMPAS dataset is a binary classification dataset that predicts whether offenders in Florida will reoffend within two years based on criminal history data. It includes 13 variables, of which 4 (number of juvenile misdemeanor, number of other convictions, number of prior adult convictions, and number of juvenile felony) are known to have monotonic increasing relationships with recidivism risk. This dataset raises ethical concerns; however, it has been used in recent publications for comparison studies and remains relevant in research fields focused on fairness.

**Blog Feedback:** The BlogFeedback dataset is a regression dataset used to predict the number of comments a blog post will receive within 24 hours. It includes 276 variables, of which 8 variables (A51, A52, A53, A54, A56, A57, A58, A59) are known to have monotonic increasing relationships with the number of comments.

**Loan Defaulter:** The Loan Defaulter dataset is a classification dataset used to predict whether a customer will default on a loan. It includes loan data from 2007 to 2015 and consists of 28 variables, among which 5 variables have shape constraints. Number of public record bankruptcies and Debt to income ratio have monotonic increasing relationships with default risk, while **Credit score**, **Length of employment**, and **Annual income** have monotonic decreasing relationships with default risk.

**Car Sales:** The Car Sales dataset is a one-dimensional regression problem aimed at predicting monthly car sales (in thousands) based on the car price (in thousands). In this problem, the **price** variable is constrained to have a convex and monotonically decreasing relationship with car sales. The dataset consists of a total of 155 entries, with 109 used for training, 32 for testing, and 14 for validation.

**Puzzle Sales:** The Puzzle Sales dataset is a regression dataset designed to predict six-month sales of wooden jigsaw puzzles using features derived from Amazon reviews. Three features are used for prediction: (1) the average star rating, which is expected to have a monotonically increasing relationship with sales; (2) *the number of reviews*; and (3) *the word count of reviews*, both of which are expected to exhibit a monotonically increasing and concave relationship with sales. The dataset includes 156 training examples, 169 validation examples, and 200 test examples.

**Wine Quality:** The Wine Quality dataset is a regression dataset designed to predict wine scores on an 80–100 scale using various wine attributes. The dataset consists of 61 variables in total: 21 binary variables representing the country of production, 39 boolean variables derived from wine descriptions published by the Wine Enthusiast Magazine, and a continuous variable representing the wine's price. Among the 120,919 data entries, 84,642 were used for training, 12,092 for validation, and 24,185 for testing. The variable *price* was included in the training process with the expectation that it has a concave and monotonically increasing relationship with wine quality.

Table 6: Descriptions for Real-world Benchmark Datasets

| Dataset | Task | # Instances | # Features | # Constrained features | Monotonic-Convex (Concave) features | Monotonic features |
|---|---|---|---|---|---|---|
| Auto-MPG | Regression | 398 | 7 | 3 | − − − − − − | **weights, displacement, horse power** |
| Blog Feedback | Regression | 54270 | 276 | 8 | − − − − − − | A51,A52,A53,A54,A56,A57,A58,A59 |
| COMPAS | Classification | 6172 | 13 | 4 | − − − − − − | number of prior adult convictions, number of juvenile felony, number of juvenile misdemeanor, number of other convictions |
| Heart Disease | Classification | 303 | 13 | 2 | − − − − − − | trestbps, chol |
| Loan Defaulter | Classification | 488909 | 28 | 5 | − − − − − − | number of public record bankruptcies, dept-to-income ratio, **credit score**, **length of employment**, **annual income** |
| Car Sales | Regression | 155 | 1 | 1 | **price** | − − − − − − |
| Puzzle Sales | Regression | 525 | 3 | 3 | *number of reviews*, *word count* | star rating |
| Wine Ratings | Regression | 120919 | 61 | 1 | *price* | − − − − − − |

### E.5 COMONET CONFIGURATION FOR HYPERBOLIC PARABOLOID FUNCTION (20)

The hyperbolic paraboloid function in equation 20 was learned using the structure in Fig. 12.

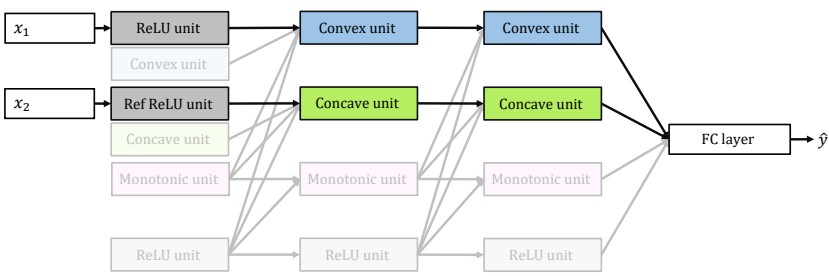

Figure 12: COMOnet Configuration for hyperbolic paraboloid function equation 20.

# F  SUPPLEMENTAL REPORT OF EXPERIMENTAL RESULTS

## F.1  TABULATED NUMERICAL RESULTS FOR FIG. 5

Fig. 5 was created using the data in the following table.

Table 7: Results of Generalization test with equation 21

| Network | Training MSE | Test MSE | Denoised test MSE |
|---|---|---|---|
| COMONet ($\lambda = 0$) | $0.01 \pm 0.00$ | $0.01 \pm 0.00$ | $0.01 \pm 0.00$ |
| COMONet ($\lambda = 1$) | $0.92 \pm 0.14$ | $0.99 \pm 0.03$ | $0.02 \pm 0.01$ |
| COMONet ($\lambda = 2$) | $3.90 \pm 0.57$ | $4.03 \pm 0.28$ | $0.06 \pm 0.02$ |
| COMONet ($\lambda = 5$) | $23.74 \pm 3.41$ | $24.88 \pm 0.89$ | $0.31 \pm 0.10$ |
| COMONet ($\lambda = 10$) | $96.84 \pm 14.66$ | $98.07 \pm 6.97$ | $0.79 \pm 0.36$ |
| COMONet ($\lambda = 20$) | $400.29 \pm 55.23$ | $402.42 \pm 5.93$ | $3.13 \pm 1.32$ |
| Same Structure ($\lambda = 0$) | $0.01 \pm 0.01$ | $0.01 \pm 0.00$ | $0.01 \pm 0.00$ |
| Same Structure ($\lambda = 1$) | $0.77 \pm 0.14$ | $1.28 \pm 0.10$ | $0.29 \pm 0.04$ |
| Same Structure ($\lambda = 2$) | $3.15 \pm 0.46$ | $5.31 \pm 0.32$ | $1.30 \pm 0.22$ |
| Same Structure ($\lambda = 5$) | $19.47 \pm 3.21$ | $32.77 \pm 1.76$ | $7.21 \pm 1.15$ |
| Same Structure ($\lambda = 10$) | $75.09 \pm 10.27$ | $128.25 \pm 6.85$ | $30.17 \pm 5.60$ |
| Same Structure ($\lambda = 20$) | $302.68 \pm 44.74$ | $510.34 \pm 21.90$ | $106.57 \pm 11.04$ |
| MLP ($\lambda = 0$) | $0.00 \pm 0.00$ | $0.00 \pm 0.00$ | $0.00 \pm 0.00$ |
| MLP ($\lambda = 1$) | $0.28 \pm 0.08$ | $1.93 \pm 0.12$ | $0.97 \pm 0.10$ |
| MLP ($\lambda = 2$) | $1.03 \pm 0.27$ | $8.15 \pm 0.59$ | $4.29 \pm 0.49$ |
| MLP ($\lambda = 5$) | $8.23 \pm 2.66$ | $48.64 \pm 4.03$ | $23.19 \pm 3.57$ |
| MLP ($\lambda = 10$) | $26.95 \pm 8.68$ | $195.11 \pm 14.12$ | $97.22 \pm 8.66$ |
| MLP ($\lambda = 20$) | $148.87 \pm 42.43$ | $722.66 \pm 61.73$ | $334.91 \pm 71.79$ |

Fig. 13 illustrates the same structure model employed in experiments on equation 21.

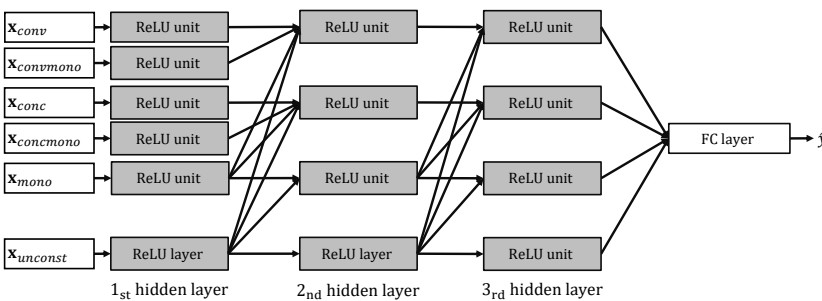

Figure 13: Same structure network

## F.2  RESULTS ON PENDER

Table 8 shows Test MSE and Convexity Score $\mathcal{C}_k$ and Monotonic Score $\mathcal{M}_k$ of PenDer (Gupta et al., 2021).

Table 8: Test MSE, Convexity Score and Monotonic Score of PenDer on Real-world Dataset.

| Dataset | Test MSE $\downarrow$ | $\mathcal{M}_k$ | $\mathcal{C}_k$ |
|---|---|---|---|
| Car Sales (conv) | $10411 \pm 107$ | 1 | 1 |
| Car Sales (conv, decr) | $10415 \pm 104$ | 1 | 1 |
| Puzzle Sales (conc) | $9428 \pm 113$ | 1 | $0.98 \pm 0.008$ |
| Puzzle Sales (conc, incr) | $9519 \pm 92$ | 1 | $0.99 \pm 0.004$ |
| Wine Quality (conc) | $5.19 \pm 0.11$ | 1 | $0.99 \pm 0.000$ |
| Wine Quality (conc, incr) | $5.27 \pm 0.20$ | $0.99 \pm 0.000$ | $0.99 \pm 0.000$ |

## F.3 HYPERBOLIC PARABOLOID WITH INTERACTION TERM

We tested whether COMONet can capture convex–concave interactions using the synthetic function

$$f(x_1, x_2) = (x_1 - 0.5)^2 - (x_2 - 0.5)^2 + \lambda\, x_1 x_2, \quad \lambda \in \{0, 1, 2, 5, 10\}.$$

Models with the interaction layer enabled (ON) were compared against models without it (OFF) using the same 1,000 samples, 80/20 split, and cross-validation protocol as in the main experiments. As shown in Table 9, the ON model maintains low error for increasing $\lambda$, while the OFF model fails to capture the interaction term.

Table 9: Interaction-layer ablation on the convex–concave synthetic function.

| Interaction | $\lambda$ | Train MSE | Test MSE |
|---|---|---|---|
| ON | 0 | $0.0001 \pm 0.0000$ | $0.0001 \pm 0.0001$ |
| ON | 1 | $0.0003 \pm 0.0011$ | $0.0004 \pm 0.0012$ |
| ON | 2 | $0.0019 \pm 0.0045$ | $0.0016 \pm 0.0045$ |
| ON | 5 | $0.0053 \pm 0.0098$ | $0.0056 \pm 0.0109$ |
| ON | 10 | $0.0209 \pm 0.0491$ | $0.0188 \pm 0.0355$ |
| OFF | 0 | $0.0001 \pm 0.0000$ | $0.0001 \pm 0.0001$ |
| OFF | 1 | $0.0080 \pm 0.0020$ | $0.0086 \pm 0.0017$ |
| OFF | 2 | $0.0318 \pm 0.0086$ | $0.0357 \pm 0.0085$ |
| OFF | 5 | $0.1951 \pm 0.0738$ | $0.2295 \pm 0.0928$ |
| OFF | 10 | $0.7576 \pm 0.2535$ | $0.8151 \pm 0.1596$ |

## F.4 FIGURES WITH THE EXPERIMENTAL RESULTS WITH (21)

Fig. 14 represent contour plots varying noise parameter $\lambda$ from 0 to 5. among 2 variables, $x_1$ is monotonic increase, and $x_2$ is convex with respect to $y$.

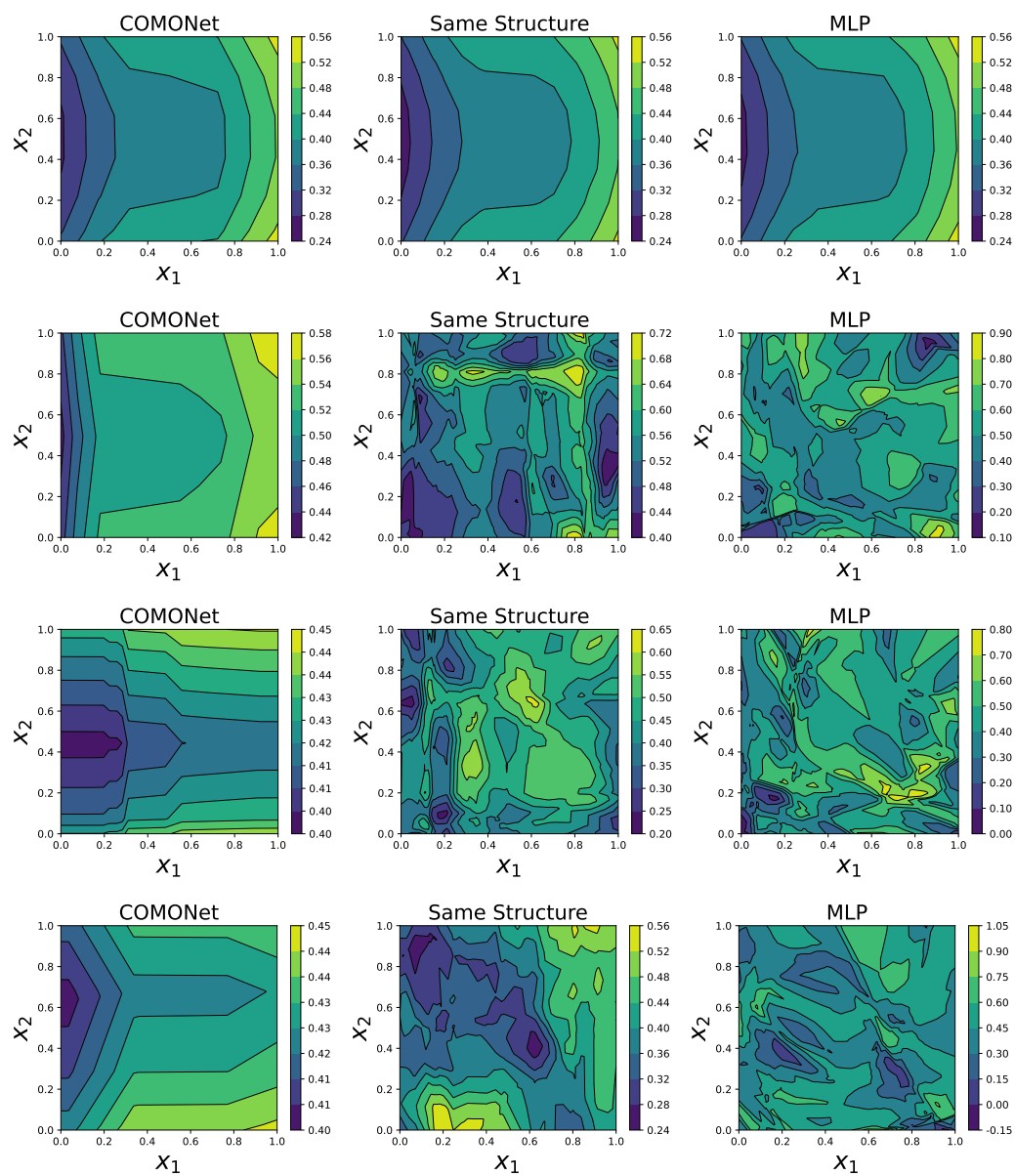

Figure 14: Contours of $x_1$ and $x_2$ varying within [0,1] when $\lambda = \{0, 1, 2, 5\}$, with $x_3$ and $x_4$ fixed at 0.5. **Left**: COMONet, **Center**: Same structure, **Right**: MLP.

Fig. 15 represent contour plots varying noise parameter $\lambda$ from 0 to 5. among 2 variables, $x_2$ is convex, and $x_3$ is monotonic-convex (increase) with respect to $y$.

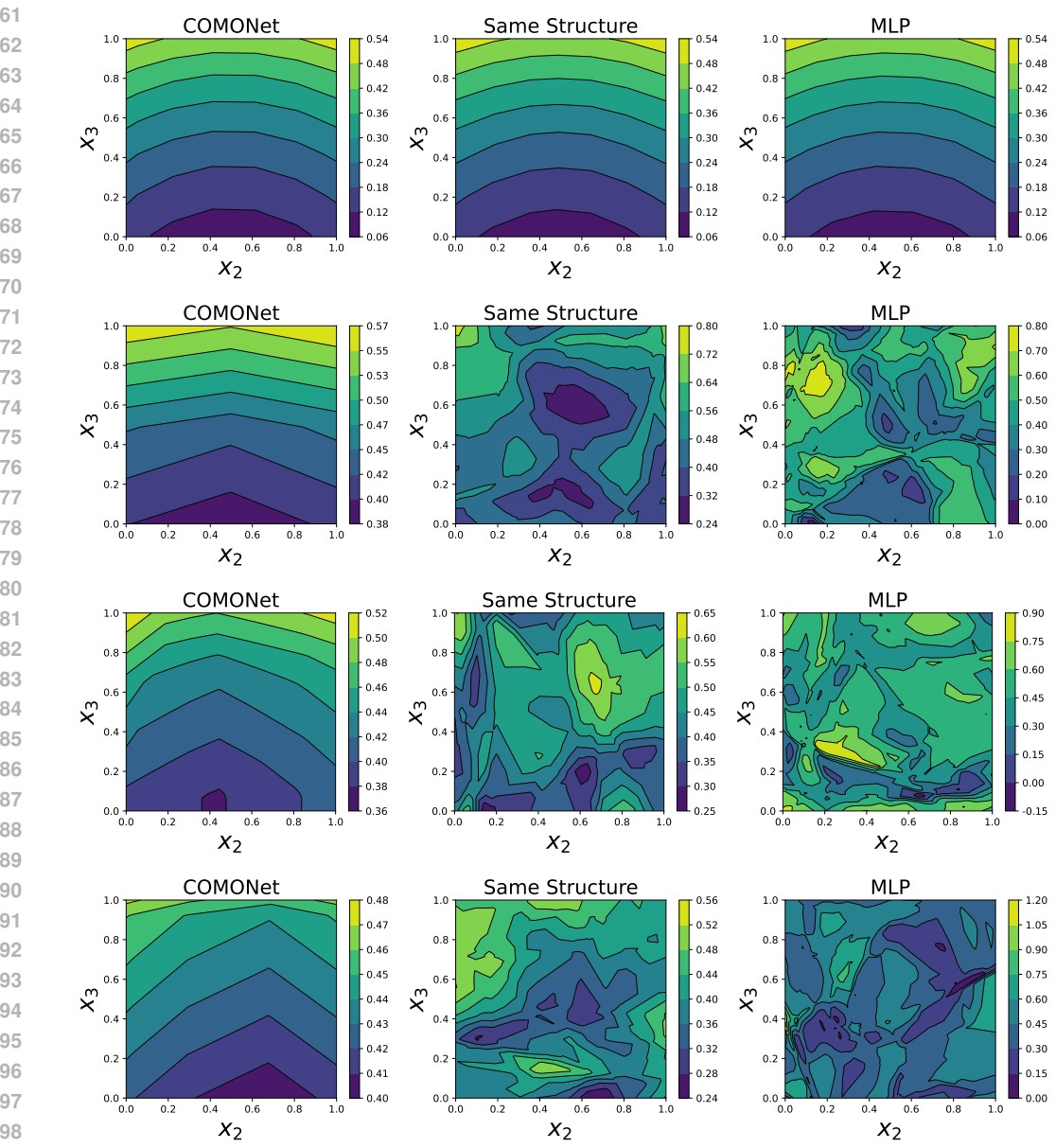

Figure 15: Contours of $x_2$ and $x_3$ varying within [0,1] when $\lambda = \{0, 1, 2, 5\}$, with $x_1$ and $x_4$ fixed at 0.5. **Left**: COMONet, **Center**: Same structure, **Right**: MLP.

# G ABLATION STUDIES

## G.1 VARIOUS ACTIVATION SETTINGS

Fig. 16 and Table 10 represent various cativation functions that satisfies conditions of Monotonicity and convexity. Among them Monotonic-convex activations can alternate ReLU and convex activations can alternate ReLU-$n$.

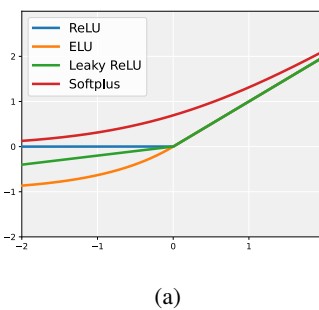 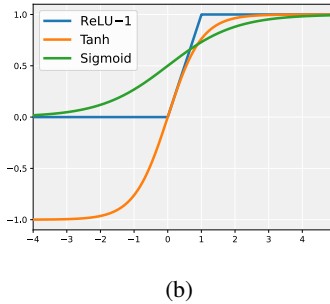

(a)          (b)

Figure 16: **Visualization of various activation functions:** activation functions in (a) is monotonic-convex function that can be used for $h_{\text{conv}}, h_{\text{relu}}, h_{\text{conc}}, h_{\text{ref-relu}}$ and (b) shows monotonic-wavy function that can be used for $h_{\text{mono}}$

Table 10: Various Activation functions

| Activation functions | Formula | Monotonicity | Convexity | Remark |
|---|---|---|---|---|
| ReLU | $\sigma(x) = \begin{cases} x & \text{if } x \geq 0, \\ 0 & \text{if } x < 0 \end{cases}$ | ✓ | ✓ | |
| Leaky ReLU | $\sigma(x) = \begin{cases} x & \text{if } x \geq 0, \\ \alpha x & \text{if } x < 0 \end{cases}$ | ✓* | ✓* | if $0 < \alpha < 1$ |
| ELU | $\sigma(x) = \begin{cases} x & \text{if } x \geq 0, \\ \alpha(e^x - 1) & \text{if } x < 0 \end{cases}$ | ✓* | ✓* | if $0 < \alpha < 1$ |
| Softplus | $\sigma(x) = \log(1 + \exp(x))$ | ✓ | ✓ | |
| Absolute | $\sigma(x) = \|x\|$ | ✗ | ✓ | unsuitable |
| ReLU-$n$ | $\sigma(x) = \begin{cases} n & \text{if } x \geq n, \\ x & \text{if } 0 \leq x < n, \\ 0 & \text{if } x < 0 \end{cases}$ | ✓ | ✗ | |
| Tanh | $\sigma(x) = \tanh(x)$ | ✓ | ✗ | |
| Sigmoid | $\sigma(x) = \frac{1}{1+e^{-x}}$ | ✓ | ✗ | |

\* Indicates conditionally achieved based on specific configurations.

Table 11 shows that no performance differences across various activations.

Table 11: Performance comparison among various activations settings († Indicates statistical ties.)

| Activation | | Auto MPG | Heart Disease | Remark |
|---|---|---|---|---|
| Monotonic activation | Convex activation | MSE ↓ | Test Acc ↑ | |
| ReLU | ReLU-1 | $7.38 \pm 1.32$† | $0.85 \pm 0.04$† | |
| ReLU | ReLU-6 | $7.38 \pm 1.32$† | $0.85 \pm 0.04$† | |
| ReLU | Sigmoid | $9.04 \pm 2.20$† | $0.84 \pm 0.05$† | |
| Leaky ReLU | Leaky relu-1 | $\mathbf{7.03 \pm 1.54}$ | $\mathbf{0.87 \pm 0.04}$ | $\alpha = 0.01$ |
| ELU | Sigmoid | $8.67 \pm 2.36$† | $\mathbf{0.87 \pm 0.04}$ | $\alpha = 0.01$ |
| ELU | Tanh | $7.20 \pm 1.59$† | $0.86 \pm 0.05$† | $\alpha = 0.01$ |
| Softplus | Sigmoid | $10.09 \pm 2.62$ | $0.84 \pm 0.06$† | |
| Softplus | Tanh | $7.46 \pm 1.57$† | $\mathbf{0.87 \pm 0.04}$ | |

### G.2 THE EFFECT OF UPWARD DIRECTIONAL CONNECTIONS

To verify whether COMONet can effectively learn interactions between variable groups, we conducted an ablation study. The function was designed as in equation 24, where the output $y$ is determined by three variables $x_1$, $x_2$ and $x_3$. Among these, $x_1$ belongs to $\mathbf{x}_{cv}$, which has a convex relationship with y, while $x_2$ belongs to $\mathbf{x}_{mn}$, which a monotonic relationship with $y$. To examine interaction effects, we varied the coefficient of interaction term $\alpha$ from 0 to 20. We generated a dataset of 1,000 samples from equation 24, using 800 for training and 200 for testing, and evaluated the models using 5-fold cross-validation repeated five times. We compared four models with different levels of connectivity in COMONet, as shown in Fig. 18: (a) Not Connected, where variable groups are completely separated; (b) Sparse-to-Specific, which allows connections only between $\mathbf{x}_{cv}$ and $\mathbf{x}_{mn}$ to facilitate learning the known interaction between $x_1$ and $x_2$ ; (c) Dense-to-Specific, where only the connection between $\mathbf{x}_{cv}$ and $\mathbf{x}_{mn}$ is removed; and (d) Fully Connected, where all groups are interconnected. As shown in Table 12 and Fig. 17, the Not Connected and Dense-to-Specific models exhibited increasing Test MSE as $\alpha$ increased, indicating a failure to capture interaction effects. In contrast, the Sparse-to-Specific and Fully Connected models maintained relatively low Test MSE despite increasing $\alpha$, demonstrating their ability to effectively learn the interaction between $x_1$ and $x_2$. These results empirically validate that COMONet can capture interactions between separated variable groups.

$$y = (x_1 - 1)^2 + \sqrt{x_2} + \alpha x_1 x_2^2 + \sin(2\pi x_3), \tag{24}$$
$$x_i \in [0, 2], \forall i \in \{1, 2, 3\},$$
$$\alpha, \in \{0, 1, 2, 5, 10, 20\}.$$

Table 12: Comparison of Test MSE Performance.

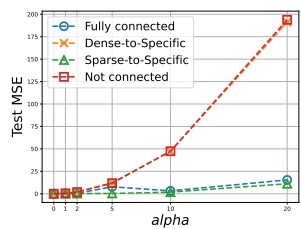

| $\alpha$ | Not connected | Sparse-to-Specific | Dense-to-Specific | Fully connected |
|---|---|---|---|---|
| $\alpha = 0$ | $0.01 \pm 0.01$ | $0.01 \pm 0.01$ | $0.02 \pm 0.00$ | $0.03 \pm 0.05$ |
| $\alpha = 1$ | $0.48 \pm 0.03$ | $0.04 \pm 0.03$ | $0.59 \pm 0.22$ | $0.49 \pm 0.20$ |
| $\alpha = 2$ | $1.87 \pm 0.11$ | $0.21 \pm 0.26$ | $1.94 \pm 0.29$ | $0.48 \pm 0.79$ |
| $\alpha = 5$ | $11.75 \pm 0.98$ | $0.33 \pm 0.08$ | $12.14 \pm 0.78$ | $7.76 \pm 6.09$ |
| $\alpha = 10$ | $47.39 \pm 6.76$ | $1.79 \pm 0.64$ | $47.20 \pm 6.56$ | $3.24 \pm 1.99$ |
| $\alpha = 20$ | $193.79 \pm 40.07$ | $11.15 \pm 12.95$ | $191.13 \pm 32.80$ | $15.41 \pm 12.44$ |

Figure 17: Visualization of The results in Table 12.

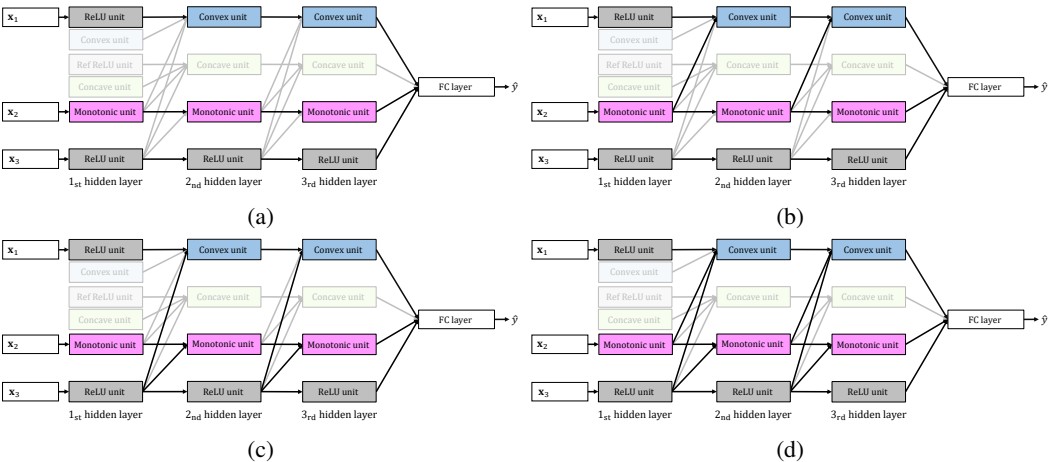

Figure 18: COMONet: (a) Not connected, (b) Sparse-to-Specific, (c) Dense-to-Specific, and (d) Fully connected.

### G.3 ABLATION STUDY ON MISASSIGNED CONSTRAINTS AND SINGLE-FEATURE LEARNING CAPABILITY

To evaluate the importance of assigning each feature from equation 21 to the correct constraint branch, we conducted an ablation where the cosine-shaped term $cos(2\pi x_4)$ was intentionally routed to incorrect branches (monotonic, convex, or monotonic–convex) instead of the intended unconstrained branch. As shown in Table 13, misassigning $x_4$ leads to substantially higher Train/Test MSE, demonstrating that each constraint branch serves a distinct purpose and that improper constraint–feature alignment severely degrades accuracy. When $x_4$ is correctly placed in the unconstrained branch, the model achieves the lowest error, confirming the necessity of an unconstrained pathway for non-monotonic and non-convex components.

Table 13: Ablation study on the assignment of $x_4$ to different constraint branches.

| Case | $x_4$ branch | Constraint | Train MSE (mean ± std) | Test MSE (mean ± std) |
|------|------|------|------|------|
| Case 1 | $x_4 \to x_m$ | monotonically increasing | $0.472 \pm 0.069$ | $0.483 \pm 0.061$ |
| Case 2 | $x_4 \to x_{mv}$ | convex & monotonically increasing | $0.468 \pm 0.094$ | $0.470 \pm 0.081$ |
| Case 3 | $x_4 \to x_{cv}$ | convex | $0.067 \pm 0.116$ | $0.063 \pm 0.108$ |
| Case 4 | $x_4 \to x_u$ | unconstrained | $\mathbf{0.004 \pm 0.003}$ | $\mathbf{0.005 \pm 0.004}$ |

We also evaluated whether each constrained component can independently model the corresponding single-feature component from equation 21. The target function was decomposed into four single-variable terms, each associated with a distinct shape constraint (monotonic, convex, monotonic–convex, and unconstrained). As shown in Table 14, training each branch on its respective term resulted in very small MSE, demonstrating that all constrained subnetworks accurately capture their intended shape behavior and reliably express the corresponding single-feature functions.

Table 14: Ablation study showing each branch's ability to independently learn its single-feature component in equation 21.

| Feature | Constraint | Train MSE (mean ± std) | Test MSE (mean ± std) |
|------|------|------|------|
| $x_1$ | monotonically increasing | $0.002 \pm 0.003$ | $0.002 \pm 0.002$ |
| $x_2$ | convex | $0.001 \pm 0.001$ | $0.001 \pm 0.001$ |
| $x_3$ | convex & monotonically increasing | $0.001 \pm 0.001$ | $0.001 \pm 0.001$ |
| $x_4$ | unconstrained | $0.002 \pm 0.002$ | $0.003 \pm 0.001$ |

## H BROADER IMPACT

Imposing shape constraints in neural networks can substantially improve model reliability, resilience, and interpretability—qualities that are especially valuable in sectors like manufacturing, finance, and healthcare where data may be limited or noisy. Nevertheless, if constraints inadvertently encode stereotypes or adverse assumptions about protected attributes (such as age, gender, or ethnicity), they risk perpetuating unfair outcomes. To guard against this, practitioners should systematically evaluate constraint behavior across different demographic groups and embed fairness checks at every stage of model development and deployment. Moreover, promoting transparency by publishing constraint definitions and associated validation tools under an open-source license fosters accountability and helps ensure these methods serve broad societal interests.

## I LLM USAGE STATEMENT

In this work, we utilized a Large Language Model (LLM) solely as an assistive tool in the writing process. The LLM was specifically employed to refine expressions and to check the clarity of mathematical formulations authored by us. Importantly, the LLM had no involvement in research ideation, the development of scientific claims, the design of experiments, or the analysis of results.

