# OpenReview forum: "A Novel Architecture for Integrating Shape Constraints in Neural Networks"
_ICLR.cc/2026/Conference — Submitted to ICLR 2026_

### Official Review · Reviewer_BzaC · 2025-10-30

**Soundness:** 1
**Presentation:** 3
**Contribution:** 1
**Rating:** 2
**Confidence:** 4

**Summary:**

This paper proposes COMONet, a neural architecture designed to enforce combinations of convexity, concavity, and monotonicity constraints by partitioning inputs into subnetworks specialized for each constraint type (convex, concave, monotonic, ReLU, and mirrored ReLU). The authors aim to guarantee constraint satisfaction by design and demonstrate empirical performance on synthetic and real-world datasets.

**Strengths:**

1) The paper is well written and attempts to generalize known apporaches to design convex neural nets.
2) The proposed model attempts to generalize existing architectures for convex and monotonic neural networks.
3) The experimental section includes both synthetic and real datasets, which helps illustrate potential applicability.

**Weaknesses:**

1. Unverified mathematical claims:
a) Equations (6)- (15) are introduced without derivation or citation to standard convex analysis texts (see Boyd & Vandenberghe, Convex Optimization, Section 3.2).
b) Theorem 4.9 is incorrect. The claim that any twice-differentiable function that is convex in one variable and concave in another must admit only a bilinear cross term is false. For instance, $f(x1,x2)=5x_1^2 - 5x_2^2+sin(x_1)sin(x_2)$ satisfies the local convex-concave property but violates the claimed decomposition. The pariwise interaction terms can be more complex than bilinear, it is however, possible that nonlinear combinations of bilinear terms can approximate such functions.
c) In equation 19, $x_1$ is not a monotonically increasing feature.

2. There is no discussion about expressivity of the proposed architecture.
Constraint enforcing architectures often trade off expressivity for guaranteed properties. The paper does not discuss whether COMONet can approximate a wide class of functions under the imposed constraints, or if the constraints severely limit the function space.

3. Lack of ablation studies.
The experimental section lacks ablation studies to isolate the contributions of different components of the architecture. As it stands, it is possible that the only reason for good empirical performance is the ReLU subnetwork, or overparameterization compared to the baselines. Ablation studies removing subnetworks or varying their sizes would help clarify this.

Overall, I do not recommend acceptance in the current form, as a lot of mathematical claims are unverified, and some are incorrect. The paper also lacks a convincing application where such a generalization of prior works that enforce convexity or monotonicity is necessary. Finally, I would also encouraage the authors to include a formal study on the exprressivity of the proposed architecture.

**Questions:**

1) Can you provide a more detailed motivation for the architectureincluding an example?

2) Can you provide proofs or references for equations 6-15?

3) Can you clarify the proof of Theorem 4.9?

4) Can you clarify the monotonicity of $x_1$ in equation 19

5) can you clarify what you mean by "The LLM was specifically employed to refine expressions and to check the clarity and
correctness of mathematical formulations authored by us."  First, I am concerned about the word "correctness".
Second, there appears to be many loose mathematical statements which are typical of chatgpt (i.e. they are not completely wrong, but not precise either).

---

> ### Author Response · Authors · 2025-11-21
> **Official Comment by Authors for Weaknesses - (1)**
>
> **Weakness1**
> - W1-(a). Please refer to our response to Question 2.
> - W1-(b). Please refer to our response to Question 3.
> - W1-(c). Please refer to our response to Question 4.
>
> **Weakness2. universial approximation theorem**
>
> We sincerely appreciate the reviewer’s insightful comment regarding the expressivity of constraint-enforcing neural architectures. We fully agree that imposing convexity, concavity, and monotonicity constraints inherently restricts the function class, and that formal expressivity analyses—such as establishing universal approximation under mixed shape constraints—constitute an important and highly challenging line of research in their own right. We also acknowledge that a thorough expressivity characterization is currently a limitation of our work. Similar to prior independent studies on the convergence or expressivity of monotone neural networks [1], we plan to address this topic as a dedicated follow-up project. This has been explicitly stated as a limitation and future direction in the revised manuscript. We kindly ask the reviewer to take this context into consideration.
>
> [1] Mikulincer, Dan, and Daniel Reichman. "Size and depth of monotone neural networks: interpolation and approximation." Advances in Neural Information Processing Systems 35 (2022): 5522-5534.

---

> ### Author Response · Authors · 2025-11-21
> **Official Comment by Authors for Weaknesses - (2)**
>
> **Weakness3. additional ablation studies.**
>
> We appreciate for insightful comment regarding the need for ablation studies. To address this, we performed additional experiments that isolate the role of each subnetwork in COMONet.
>
> We appreciate the reviewer’s question regarding the necessity of the unconstrained component and have added an experiment to aid readers’ understanding. In Eq. (21) in rivisited PDF, the variable $x_4$ is unconstrained and follows a cosine-shaped target, $\cos(2\pi x_4)$, with $x_4\in [0,1]$. In our original setup, because $x_4 \in \mathcal{U}$, it is routed to the bottom ReLU unit in Fig. 2.
> In the new ablation, we reassign $x_4$ to each of $\mathcal{M}$, $\mathcal{CV}$, and $\mathcal{MV}$ in turn and train the model under those constraints. The results are summarized below.
>
> Findings:
>
> 1. Under monotonic constraints (Case 1 and 2), the model cannot approximate the non-monotone cosine shape of $x_4$, leading to substantially higher MSE.
> 2. Under a convex constraint (Case 3), the MSE is lower than in the monotone cases, suggesting that a quadratic-like convex function can partially mimic segments of $\cos(2\pi x_4)$ over limited intervals $x_4\in[0,1]$, yet it still fails to capture the full oscillatory pattern.
> 3. When $x_4$ is correctly placed in $\mathcal{U}$ (Case 4), the model achieves the lowest MSE, showing that the unconstrained branch is essential for accurately representing variables with no monotonic or convex structure.
>
> | Case | $x_4$ branch | Constraint | Train MSE (mean ± std) | Test MSE (mean ± std) |
> |------|----------------|------------|-------------------------|------------------------|
> | Case 1 | $x_4 \to x_m$ | monotonically increasing | 0.472 ± 0.069 | 0.483 ± 0.061 |
> | Case 2 | $x_4 \to x_{mv}$ | convex & monotonically increasing | 0.468 ± 0.094 | 0.470 ± 0.081 |
> | Case 3 | $x_4 \to x_{cv}$ | convex | 0.067 ± 0.116 | 0.063 ± 0.108 |
> | Case 4 | $x_4 \to x_u$ | unconstrained | **0.004 ± 0.003** | **0.005 ± 0.004** |
>
> We also conducted an ablation study to examine whether each component can accurately learn the single-feature function associated with its assigned constraint.
> To this end, we tested whether each constrained branch can explicitly model its corresponding term in the target function. Using Eq. (21) without the noise term, the target decomposes into four single-feature components:
>
> - $x_1$: monotonically increasing, wavy-yet-monotone
> - $x_2$: convex
> - $x_3$: both monotonically increasing and convex
> - $x_4$: unconstrained (cosine)
>
> We then verified, term by term, that each branch dedicated to a given feature can accurately represent its corresponding single-feature component. Experiments were conducted in the noise-free setting, and the per-term MSEs were small.
>
> | Feature | Constraint | Train MSE (mean ± std) | Test MSE (mean ± std) |
> |---------|------------|-------------------------|------------------------|
> | $x_1$ | monotonically increasing | 0.002 ± 0.003 | 0.002 ± 0.002 |
> | $x_2$ | convex | 0.001 ± 0.001 | 0.001 ± 0.001 |
> | $x_3$ | convex & monotonically increasing | 0.001 ± 0.001 | 0.001 ± 0.001 |
> | $x_4$ | unconstrained | 0.002 ± 0.002 | 0.003 ± 0.001 |
>
> These two ablation studies allow us to clearly isolate and verify how each component of the proposed method contributes to the overall performance.
>
> In addition, to address your concern and ensure a fair comparison with benchmark methods, we conducted further experiments using a reduced number of parameters in our model. The corresponding results are provided below.
>
> | Model | Dataset | Constraint | # parameters | Test MSE (mean ± std) |
> |---------|---------|------------|-------------------------|------------------------|
> | COMONet | Carsales | (conc) | 1195 | 10391 ± 140 |
> | COMONet | Carsales | (conc, incr) | 1195 | 10410 ± 128 |
> | COMONet | Puzzle Sales | (conc) | 1819 | 9409 ± 41 |
> | COMONet | Puzzle Sales | (conc, incr) | 1819 | 9263 ± 86 |
>
> These results show that the proposed method achieves performance comparable to the baseline methods even when using fewer parameters.
> We have incorporated these results into the revised manuscript, specifically in Section 5.2 (Table 3), Appendix E.1 (Table 4), and Appendix G.3. We kindly invite the reviewer to refer to these sections for further details.
> We hope that the additional ablation studies and the experiments conducted with a reduced parameter budget help alleviate your concerns.

---

> ### Author Response · Authors · 2025-11-21
> **Official Comment by Authors for Questions - (1)**
>
> **Question1. detailed motivation for the architecture**
> Thank you for requesting a more detailed motivation for our architecture. Our goal is to develop a unified neural network that can simultaneously satisfy multiple shape constraints—convexity, concavity, and monotonicity—while also capturing interactions across constraint-specific variable groups. Although prior work enforces convex/concave or monotonicity constraints individually, no existing architecture can jointly guarantee strict constraint satisfaction and model cross-group interactions. This gap is critical in real-world settings where mixed constraints and interaction terms naturally arise. To illustrate the necessity of our design, we conducted two additional experiments.
>
> (1) Hyperbolic paraboloid + interaction term: We constructed a synthetic equation (20) that exhibits both convex and concave directions, forming a saddle structure, and further added an explicit interaction term.
>
> $$
> f(x_1, x_2) = (x_1 - 0.5)^2 - (x_2 - 0.5)^2 + \lambda \cdot x_1 \cdot x_2
> $$
>
> We varied the interaction strength $\lambda$ across $\{0, 1, 2, 5, 10\}$ and compared COMONet with the interaction layer enabled (ON) and disabled (OFF), using identical training protocols.
> We sampled 1,000 points uniformly from $[0,1]^2$, used an 80/20 train/test split, and followed the paper’s protocol of 5-fold CV repeated 5 times (total 25 runs).  We report mean ± std across runs.
> As $\lambda$ increases, the interaction-ON model consistently achieves lower test error than the interaction-OFF model, aligning with our theoretical result that the layer captures the permissible convex–concave cross-term.
> (All runs followed the same constraint settings as in the main experiments.)
>
> | interaction layer | $\lambda$ | Train MSE (mean ± std) | Test MSE (mean ± std) |
> |-------------------|------------:|-------------------------|------------------------|
> | ON  | 0  | 0.0001 ± 0.0000 | 0.0001 ± 0.0001 |
> | ON  | 1  | 0.0003 ± 0.0011 | 0.0004 ± 0.0012 |
> | ON  | 2  | 0.0019 ± 0.0045 | 0.0016 ± 0.0045 |
> | ON  | 5  | 0.0053 ± 0.0098 | 0.0056 ± 0.0109 |
> | ON  | 10 | 0.0209 ± 0.0491 | 0.0188 ± 0.0355 |
> | OFF | 0  | 0.0001 ± 0.0000 | 0.0001 ± 0.0001 |
> | OFF | 1  | 0.0080 ± 0.0020 | 0.0086 ± 0.0017 |
> | OFF | 2  | 0.0318 ± 0.0086 | 0.0357 ± 0.0085 |
> | OFF | 5  | 0.1951 ± 0.0738 | 0.2295 ± 0.0928 |
> | OFF | 10 | 0.7576 ± 0.2535 | 0.8151 ± 0.1596 |
>
> (2) Aggregation baseline comparison: Using the synthetic function from Appendix G.2, we compared COMONet against a naive aggregation of SCNN (for $x_v$) and SMNN (for $x_m$). As the interaction strength increased, the aggregated SCNN+SMNN model exhibited rapidly growing error—indicating its inability to represent variable-group interactions—whereas COMONet maintained low error across all settings. This empirically supports that interaction-awareness is essential and cannot be achieved through simple additive combinations of existing constrained networks.
>
> $$
> y = (x_1 - 1)^2 + \sqrt{x_2} + \alpha x_1 x_2^2 + \sin(2\pi x_3)
> $$
> $$
> x_i \in [0,2], \; \forall i \in \{1,2,3\}
> $$
> $$
> \alpha \in \{0,1,2,5,10,20\}
> $$
>
> This dataset introduces controlled interactions between $x_v$ and $x_m$, and we evaluated performance while gradually increasing the interaction strength $\alpha$.
>
> - $x_1$: convex feature $x_v$ (since $\frac{\partial^2 y}{\partial x_1^2} = 2$)
> - $x_2$: monotonic increasing feature $x_m$ (since $\frac{\partial y}{\partial x_2} = \frac{1}{2\sqrt{x_2}} + 2\alpha x_1 x_2$)
>
> We implemented the aggregation of SCNN and SMNN as defined in Equation (24):
> - $x_1 (x_v)$ was fed into the SCNN branch
> - $x_2 (x_m)$ and $x_3 (x_u)$ were fed into the SMNN branch
> Both branches were trained jointly. The experimental setup follows Appendix G.2. Results are shown below:
>
> | alpha | Test MSE (SCNN+SMNN) | Test MSE (COMONet) |
> |------|-----------------------|---------------------|
> | $\alpha$ = 0 | 0.01 ± 0.00 | 0.03 ± 0.05 |
> | $\alpha$ = 1 | 0.53 ± 0.13 | 0.49 ± 0.20 |
> | $\alpha$ = 2 | 1.79 ± 0.82 | 0.48 ± 0.79 |
> | $\alpha$ = 5 | 12.65 ± 6.05 | 7.76 ± 6.09 |
> | $\alpha$ = 10 | 49.37 ± 6.99 | 3.24 ± 1.99 |
> | $\alpha$ = 20 | 192.25 ± 37.93 | 15.41 ± 12.44 |
>
> These results reinforce the motivation for our architecture: a single model is required to jointly enforce heterogeneous shape constraints while structurally capturing interactions, a capability absent in existing methods. In addition, we have added several real-world examples in the Introduction—such as HVAC control, physical modeling, and economic utility functions—where convex, concave, and monotonic behaviors coexist. For further examples and discussion, we kindly refer the reviewer to our response to Reviewer 3grh.

---

> ### Author Response · Authors · 2025-11-21
> **Official Comment by Authors for Questions - (2)**
>
> **Question2. Equations (6)–(15)**
>
> - Q2. We appreciate the reviewer’s careful reading. Equations (6)–(15) provide a formalized description of the architecture in Fig. 2, specifying how the five types of subnetworks (convex, concave, monotonic, ReLU, and mirrored-ReLU) are combined to construct each hidden layer. These equations do not assert any new convex-analytic property by themselves; they only encode the computational graph of COMONet in a compact mathematical form. All theoretical guarantees regarding convexity, concavity, and monotonicity of the composed network rather follow from:  Lemma 1–5, which state the curvature/monotonicity properties of each unit type, and Theorems 4.6–4.8, which prove that the overall architecture satisfies the intended shape constraints. The full proofs of these lemmas and theorems are provided in Appendix A. To avoid confusion and improve clarity, we have revised the descriptions in the Network Structure section.
>
> **Question3. Theorem 4.9**
>
> We sincerely thank the reviewer for providing the counterexample
> $$
> f(x_1, x_2) = 5x_1^2 - 5x_2^2 + \sin(x_1)\sin(x_2),
> $$
> which indeed satisfies the convex–concave structure but does not admit the bilinear decomposition originally stated in Theorem 4.9. This was an excellent observation, and we appreciate the reviewer’s careful analysis. It identified an important issue in our previous formulation and significantly improved the rigor of our paper.
>
> To address this, we have **revised Theorem 4.9**. The revised theorem does *not* claim that all convex–concave functions must reduce to a bilinear interaction. Instead, it establishes the following:
>
> When convexity in \(x_1\) and concavity in \(x_2\) must be preserved **for all admissible single-variable curvature functions**
> $$
> g''(x_1) \ge 0, \qquad h''(x_2) \le 0,
> $$
> the **only interaction term** that can *never* violate these curvature assignments—regardless of the dataset or the learned magnitudes of curvature—is the bilinear term
> $$
> \phi(x_1, x_2) = \alpha x_1 x_2, \qquad \alpha \in \mathbb{R}.
> $$
>
> This clarification is crucial because COMONet must ensure that convexity and concavity constraints are satisfied **under all possible learned curvature values**, not just for a specific target function. Therefore, while the interaction layer is not designed to approximate every possible convex–concave interaction, it provides a **structurally safe interaction mechanism** that guarantees curvature preservation independently of the data.
>
> In summary, the revised theorem emphasizes that the bilinear interaction is the **curvature-preserving choice** for COMONet’s setting, which prioritizes structural constraints over unrestricted expressivity.
> We are genuinely grateful to the reviewer for this constructive feedback.
> It substantially improved the mathematical correctness and clarity of our theoretical formulation.
>
>
> **Question4. monotonicity of $x_1$ in  Eq. (19)**
> We thank the reviewer for the helpful comment. Before clarifying the monotonicity of Equation (19) (Equation (21) in revision), we have revised the formula by adding explicit parentheses around the argument of the sine function to avoid any notational ambiguity. The monotonicity of Equation (19) follows directly from its derivative. For the function $f(x_1) = \frac{2\pi{x_1}+\sin{(2\pi{x_1})}}{2\pi}$, we have $f'(x_1) = \frac{2\pi+2\pi\cos{(2\pi{x_1})}}{2\pi}$ $ =1 + \cos{(2\pi{x_1})}\ge0$. Thus the function is globally monotone increasing for $x_1$.
> Additionally, if the concern arises from interpreting “increasing” as “strictly increasing,” we note that on Page 1 we explicitly state that, for readability, we refer to non-decreasing functions as increasing throughout the paper. This convention ensures that points where the derivative becomes zero do not violate the intended notion of monotonicity.
>
> **Question5. LLM USAGE STATEMENT**
>
> We appreciate the reviewer’s concern regarding the use of the term “correctness.”
> Our work defines multiple variable groups and unit types, each with its own notation and indexing rules. To avoid potential confusion arising from notational inconsistencies, we used an LLM only to check the consistency and clarity of expressions during writing.
>
> We hope this clarification addresses the reviewer’s concern. Accordingly, we have updated our LLM Usage Statement as follows:
>
> “In this work, a Large Language Model (LLM) was used solely as an assistive tool during the writing process, specifically to refine phrasing and verify the clarity and consistency of mathematical expressions written by the authors. The LLM played no role in research ideation, theoretical development, experimental design, or analysis.”
>
> We sincerely appreciate your thoughtful feedback. Your comments have been invaluable in improving the quality and clarity of our manuscript throughout the review process. We hope that the revised submission, together with our rebuttal, adequately addresses your concerns.
>
> — The authors

---

### Official Review · Reviewer_1yni · 2025-10-31

**Soundness:** 3
**Presentation:** 4
**Contribution:** 3
**Rating:** 4
**Confidence:** 5

**Summary:**

Authors propose a new network architecture that uses partial connections impose monotonicity and convexity/concavity shape constraints. Experimental results are good but not impressive.

**Strengths:**

This is a practical problem that warrants more research, and their approach seems reasonable and not overly complex.

Nice intro, but I highly recommend also including a simple real-work example in the intro to help readers grasp the issue and why you might want a constraint like this.

On Motivation: Shape constraints are a terrific regularizer, and they aren’t just for specialized needs like healthcare, shape constraints (via the lattice models approach of Gupta et al.) have been widely used at Google in all sorts of ML just for the regularization, interpretability, and reduced churn, there are some great real-world case studies in their 2016 JMLR paper, Monotonic Calibrated Interpolated Look-Up Tables as well as some of the experiments in their later shape constraint papers (some of which you already cite).  Shape constraints are also useful for for fairness (Wang, AI Stats 2020)

I thought the coverage of related work was good (but the details about the related work were not suffciently accurate - see Weaknesses)

The visualization design of Fig 1 was great (but not sufficiently accurate - see Weaknesses).

**Weaknesses:**

It is a good paper, the main negative is just that the proposal doesn't seem super significant in terms of what they propose or the experimental achievements, and the bar for NeurIPS is just so high that I'd have trouble championing this paper to get in, but I definitely think there's some nice work here and would like to see it published somewhere.

I have some kvetching about their description of the related work in lattice models, which have been widely used at Google for shape-constrained ML.

Re: Related work - Authors say “PenDer (Gupta et al., 2021)
appears to be the only existing method capable of incorporating all eight shape constraints depicted in
Fig. 1(a).” but Google’s lattice models can incorporate all of those constraints as well.  In fact the Gupta 2018 paper you cite says in the abstract “We show that one can build flexible, nonlinear, multi-dimensional models using lattice functions with any combination of concavity/convexity and monotonicity constraints on any subsets of features”. It is incorrect that your chart in Fig 1 b has X’s (no’s) for Shape-constrained Lattice for only-monotonic increasing and monotonic decreasing, but in fact that’s a standard use case of shape-constrained lattices, see the 2016 JMLR paper
Monotonic Calibrated Interpolated Look-Up Tables.  However, Table Fig 1.b is is correct for SCNN.

re: “Due to this limitation, there remains a need for
the development of a method that can integrate all possible shape constraints while strictly guarantee
their satisfaction.”  Lattice models can do that.

Apart from thinking you are incorrect about what can be done with Lattice models, I really liked the information presentation of Fig. 1, thanks!

Re: “However, the former relies on a lattice structure to enforce shape constraints,
resulting in significantly increased computational complexity as the input dimension grows, making it non-scalable.”  While that’s true of a single lattice, note that the papers you cite use an ensemble of such lattices to handle large numbers of features and as their papers prove, the ensemble satisfies these shape constraints as long as the base models do. So it’s not reasonable to say that work doesn’t scale when it does.

Re: “We introduce three
classes of local shape constraints—partial convexity, partial concavity and partial monotonixity—that
apply to subsets of the input coordinates.”  You aren’t “introducing” any of those, those are all known concepts (and typo in monotonicity).

MINOR:
- Seems weird to cite a 2010 paper for ReLU, I mean they date to Householder’s 1941 work, or maybe cite some later paper that digs into them in a more neural-nety usage but that would be some paper from the 60’s, or citing any ML textbook is fine, but also I don’t think one needs to cite anything for standard ReLU at this point.

- Use {} around titles in your bib tex to get them to capitalize correctly and consistently please.

**Questions:**

For multiple inputs, do they handle/impose joint convexity (like Amos) or ceterus paribus convexity like lattice models (see the discussion and examples on this in Section 1 of Diminishing Returns Shape Constraints, Gupta et al. 2018). Your definition of partial convexity looks like a ceterus paribus definition, but your architecture is so similar to SCNN that I think you will get joint convexity (which is a more restrictive constraint).

Is the proposal to use RELU units necessary, that is, is that choice that key to achieving the shape constraints, or would any convex activation function work fine?

In Table 3 the Puzzle Sales Test MSE for SCNN concave and increasing is listed as 9258 +/319, but in the Gupta et al. 2018 Diminishing Returns paper they give it the much better value of Test MSE: 6927, and the dataset has a fixed, real-world non-iid Train/Val/Test split. Did you use this dataset differently and that’s why the results are different? Can you provide the results for their real-world non-iid Train/Val/Test split?

---

> ### Author Response · Authors · 2025-11-21
> **Official Comment by Authors for Strengths, Weaknesses and Minor**
>
> **Real-World Examples and Contributions of the Paper**
>
> Thank you for your thoughtful and constructive review. We especially appreciate your comments highlighting that shape constraints serve as powerful regularizers and are broadly useful beyond specialized domains such as healthcare, enginnering.
>
> We have added real-world examples to the introduction, including cases from HVAC control, portfolio management, and physics, where combinations of monotonic, convex, and concave constraints naturally arise in actual systems (Please refer to the section (2) On real-world applications requiring mixed constraints in our rebuttal to Reviewer 3grh for a more detailed explanation.). We have also incorporated the fairness-related reference you mentioned (Wang, AISTATS, 2020). We believe these additions help clarify the motivation and contributions of the paper more effectively.
>
> **Regarding Shape-Constrained Lattice Models**
>
> We sincerely appreciate the reviewer’s detailed and insightful feedback regarding Shape-Constrained Lattice models. We carefully reviewed all points and revised the manuscript accordingly, as summarized below.
>
> (a) All combinations of shape constraints are supported by lattice models
>
> As the reviewer correctly pointed out, Gupta et al. (2018) clearly state in both the abstract and main text that lattice models can incorporate any combination of monotonicity, convexity, and concavity constraints.
>
> In Fig. 1(b) of our original submission, several shape constraints (monotonic increasing (or decreasing)) were mistakenly marked as unsupported for Shape-Constrained Lattice models. This was an oversight on our part, and we have updated the figure to accurately reflect the full range of shape constraints supported by lattice models based on the reviewer’s comment.
>
> (b) Practical limitations of RTL in enforcing convexity
>
> We additionally verified that while RTL is structurally capable of representing convex constraints, the number of inequalities required to enforce convexity becomes extremely large in practice. As a result, RTL relies on stochastic projection steps (Light touch algorithm), which may lead to occasional violations of the convexity constraints during actual training.
>
> To clarify this distinction, Fig. 1(b) now differentiates between structural guarantees and practical guarantees (*), and we have expanded the corresponding explanation in the main text.
>
> (c) Revision of the statement regarding the need for proposed method
>
> As the reviewer noted, lattice models are also capable of providing strict guarantees. Therefore, the previous statement suggesting the need for a “new method that can strictly guarantee all shape constraints” was inaccurate. We have revised the discussion in paragraph (Shape-Constrained Neural Networks) in section 2 to accurately reflect the capabilities of existing methods and to correct any misleading implications.
>
> All revisions related to SCLattice can be found in the updated manuscript, specifically in Section 1, Section 2, and Figure 1. We kindly invite you to refer to these parts for full details.
>
> **Minor**
>
> Thank you for these minor suggestions.
> (1) We agree that citing ReLU is unnecessary, and we have removed the citation.
> (2) We also corrected the capitalization issues by enclosing the relevant BibTeX titles in braces.

---

> ### Author Response · Authors · 2025-11-21
> **Official Comment by Authors for Questions**
>
> **Question 1**
>
> Thank you for raising this important point. We clarify that our definition indeed corresponds to joint convexity (and concavity) on the variable block $\mathbf{x}_v$, not coordinate-wise (ceteris paribus) convexity. In our formulation, $\mathbf{x}_v \in \mathbb{R}^{|\mathcal{v}|}$ is treated as a vector block, and the convex combination is taken jointly over all coordinates in that block.Thus, the constraint exactly matches the joint convexity imposed in ICNN-style architectures. To avoid confusion, we have added an explicit clarification in the revised manuscript that the definition and architecture enforce joint partial convexity/concavity.
>
> As you correctly pointed out, joint convexity (or concavity) is indeed a much stronger form of curvature constraint. However, for certain downstream tasks—such as optimization or optimal input search—these stronger curvature properties can be highly desirable or even necessary.
>
> At the same time, joint curvature constraints may become overly restrictive in practical learning scenarios. To relax these constraints and allow more flexible, variable-wise curvature behavior, the revised manuscript introduces a bilinear intra-interaction term between jointly convex (or jointly concave) variable groups. This formulation is formally presented in **Theorem 4.10**. This interaction term can be applied optionally, and we show that it can enhance the expressive capacity of the model when additional flexibility is beneficial.
>
> **Question 2**
>
> Yes, any activation function that is both convex and monotonic can be used in place of ReLU. ReLU is simply one convenient choice, not a requirement for enforcing the shape constraints. In fact, Appendix G.1 includes experimental results demonstrating that our method works with various activation functions.
> e. g., ELU, Softplus, and other convex and monotonic activations. We kindly refer the reviewer to Appendix G.1 for further details.
>
> **Question 3**
>
> Thank you for pointing this out — we agree that this is an important detail worth clarifying.
> The Puzzle Sales dataset does indeed come with a fixed, real-world non-IID Train/Val/Test split, and all of our experiments strictly follow this original split.
>
> During reproduction, however, we were unable to match the SCNN test performance reported in Gupta et al. (2018), despite our best efforts. To ensure fairness and accuracy, we carefully re-implemented SCNN (the implementation is provided in the Supplementary Material) and verified the results once more. The numbers reported in Table 3 are therefore based on the exact real-world non-IID Train/Val/Test split provided with the dataset.
>
> Your review was very helpful in improving our manuscript during the rebuttal period. We sincerely appreciate your constructive feedback.
> — The authors

---

### Official Review · Reviewer_3grh · 2025-10-31

**Soundness:** 4
**Presentation:** 4
**Contribution:** 2
**Rating:** 6
**Confidence:** 4

**Summary:**

This paper introduces a new architecture that is able to enforce multiple shape constraints simultaneously, such as monotonicity, convexity, concavity, and their combinations. Existing methods either support only a subset of these constraints or do not guarantee compliance. On the other hand, the proposed approach is able to enforce shape constraint types at the same time and offers theoretical guarantees.
This is achieved by routing the features through modules with specific behavior, masking the connectivity of the network units to maintain the desired shape constraints.
The method is evalutaed both on synthetic and real-world datasets showing better or comparable predictive performance compared to state-of-the-art.

**Strengths:**

The paper is well structured and very clear. The idea is interesting and well executed. The experimental evaluation covers its basis and seems convincing and exhaustive.

**Weaknesses:**

The proposed network mostly feels like a combination of previous architectures, which makes the novelty of the contribution a bit limited. Also, I am unsure that the combination of multiple shape constraints in a single interconnected network is particularly useful in itself.

**Questions:**

Can you provide more concrete examples of application where satisfying multiple constrains simultaneously is critical?

---

> ### Author Response · Authors · 2025-11-21
>
> We sincerely thank you for the constructive and thoughtful feedback.
>
> **(1) contributions and novelty**
>
> While COMONet reuses known primitive modules (monotonic layers, ICNN layers, etc.), its contribution is not simply a “combination” of existing architectures.
> To the best of our knowledge, COMONet is the only model that can enforce multiple shape constraints simultaneously—and under strict guarantees—even in the presence of interactions among variables.
>
> For a more detailed explanation, we kindly invite the reviewer to refer to the Weaknesses section of the response to reviewer pS2q.
>
> **(2) On real-world applications requiring mixed constraints**
>
> Regarding the usefulness of combining multiple shape constraints, we note that other reviewers independently emphasized exactly this need. Reviewer pS2q pointed out that in real applications “a single feature may exhibit different shape behaviors depending on its interaction with other features,” directly motivating architectures that can jointly model convex, concave, and interacting variables. Reviewer 1yni further highlighted that shape constraints are “a terrific regularizer” widely used in industry for stability, interpretability, and reduced churn. These comments confirm both the practical relevance and the value of extending shape-constrained modeling beyond single-constraint architectures. COMONet aims precisely to fill this gap.
>
> There are several domains where different input variables obey different shape behaviors simultaneously.
>
> - exmaple 1. mean–variance utility introduced by Markowitz (1952):
>
> $$
> \max_{w} \; U(w)
> = w^{\top}\mu \;-\; \lambda\, w^{\top}\Sigma w .
> $$
> Here, $w^{\top}\mu$ is linear, while $w^{\top}\Sigma w$ is convex, so $-\lambda\, w^{\top}\Sigma w$ becomes concave. Thus the objective mixes linear and convex--concave terms within a single function, revealing heterogeneous curvature across variables.
>
> - example 2. Heating, Ventilating, Air Conditioning(HVAC) system :
>
> In HVAC systems, different variables naturally exhibit heterogeneous curvature. Fan power increases in a **convex** manner with airflow, because energy consumption grows disproportionately at higher speeds. In contrast, thermal comfort penalties are typically **concave** in the supply temperature, reflecting diminishing sensitivity as the temperature moves away from the comfort point.
> When airflow is adjusted as a function of temperature, these components interact, producing a mixed convex–concave structure within a single system.
>
>
> **Question**
>
> Please refer to our response in **(2) On real-world applications requiring mixed constraints**, where we provide concrete examples and justification for the practical necessity of mixed shape behaviors in real systems.
>
>
>
> We hope that our responses adequately address your concerns regarding the novelty and contribution of our work. Thank you sincerely for your constructive review.
> — The authors

---

> > ### Comment · Reviewer_3grh · 2025-11-21
> >
> > First of all, I really appreciated the authors efforts put into the revision.
> >
> > __Reply to (1)__
> >
> > I understand that COMONet is able to enforce multiple shape constraint at the same time. This is achieved by imposing a specific subnetwork structure that allows for the constraints to be preserved. The individual building blocks are very similar to previous works and the masking of connections is also present in previous literature. It seems to me that the main novelty is simply the combination of convex/concave units and monotonic ones in the same network.
> >
> > __Reply to (2)__
> >
> > I believe the examples of application of the joint mixed constraint given are a bit of a stretch. I fail to see how a NN would be needed/useful in the first example, while the second only uses convex/concave constraints and does not mix in monotonicity.

---

> > > ### Author Response · Authors · 2025-11-24
> > >
> > > We apologize for not fully addressing your concerns in our earlier response. We have additionally provided an example in which convex, concave, and monotonic relationships all arise simultaneously.
> > >
> > > **Asset pricing Setting:**
> > >
> > > We assume the following well-established relationships between asset returns and key accounting variables:
> > >
> > > - R&D investment ratio ($x_1$)  → returns ($y$) : concave, monotonic increasing. Higher R&D intensity is generally associated with higher expected returns, while exhibiting diminishing marginal effects at elevated levels of investment.
> > >
> > > - Illiquidity ($x_2$) → returns ($y$) : monotonic increasing. Lower liquidity requires higher compensation from investors, leading to higher expected returns (the illiquidity premium).
> > >
> > > - Earning ($x_3$), cash Flows ($x_4$)-> returns ($y$) : convex, monotonic increasing. Firms with stronger earnings or cash flows tend to exhibit disproportionately higher future expected returns.
> > >
> > > [1] Gu, Shihao, Bryan Kelly, and Dacheng Xiu. "Empirical asset pricing via machine learning." The Review of Financial Studies 33.5 (2020): 2223-2273.
> > >
> > > [2] Fama, Eugene F., and Kenneth R. French. "A five-factor asset pricing model." Journal of financial economics 116.1 (2015): 1-22.
> > >
> > > [3] Breuer, Matthias, and David Windisch. "Investment Dynamics and Earnings‐Return Properties: A Structural Approach." Journal of Accounting Research 57.3 (2019): 639-674.
> > >
> > > [4] Billings, Mary Brooke, Matthew C. Cedergren, and Stephen G. Ryan. "Continuation Options and Returns-Earnings Convexity." Available at SSRN 1945908 (2011).
> > >
> > > Notably, as demonstrated in [1], neural networks exhibit clear advantages in asset-pricing tasks by capturing nonlinear predictive structures and high-order interactions, and the paper further concludes that shallow architectures perform best under noisy and data-limited financial settings—conditions under which our proposed method is particularly effective—so we will incorporate these supporting citations directly into the introduction to clarify the motivation.
> > >
> > >
> > > **Concrete strength problem:**
> > >
> > > In concrete strength prediction, the relationships between the compressive strength and the following input variables exhibit the patterns described below.
> > >
> > > - Cement content ($x_1$) → concrete strength ($y$): monotone increasing, concave (diminishing returns)
> > > - Water–cement ratio ($x_2$) → concrete strength ($y$): strongly monotone decreasing (Abrams’ law)
> > > - Fine aggregate ratio ($x_3$) → concrete strength ($y$): convex (U-shaped) both too little and too much reduce strength
> > > - Admixture dosage ($x_4$) → concrete strength ($y$): concave due to saturation effects
> > > - Age ($x_5$) → concrete strength ($y$): convex, monotone increasing (diminishing returns)
> > >
> > > Thus, even in a well-studied engineering domain such as concrete, we observe that convex, concave, and monotonic relationships can coexist with respect to the same output $y$.
> > >
> > > [5] Yeh, I-Cheng. "Generalization of strength versus water–cementitious ratio relationship to age." Cement and Concrete Research 36.10 (2006): 1865-1873.
> > >
> > > **Robot dynamics:**
> > >
> > > Classical robot dynamics further demonstrate that a single physical system can simultaneously exhibit convex, concave, and monotonic relationships.
> > >
> > > $\tau$ → Torque contribution:
> > >
> > > - kinetic energy term -> $\tau$ : convex , monotone increasing
> > > - Friction torque -> $\tau$ : concave , monotone decreasing
> > > - Gravity term -> $\tau$ : monotone increasing (within operation range)
> > >
> > > [6] Spong, Mark W., Seth Hutchinson, and Mathukumalli Vidyasagar. Robot modeling and control. Vol. 3. New York: Wiley, 2006.
> > >
> > > [7] Armstrong-Hélouvry, Brian, Pierre Dupont, and Carlos Canudas De Wit. "A survey of models, analysis tools and compensation methods for the control of machines with friction." Automatica 30.7 (1994): 1083-1138.
> > >
> > > **HVAC:**
> > >
> > > We would also like to note that certain HVAC variables also exhibit monotonic behavior. For instance, Compressor RPM and discharge temperature follow a strong monotonic-increasing relationship, which is a standard characteristic of vapor-compression cycles.
> > >
> > > We hope that these additional examples and clarifications help address your concerns. We sincerely appreciate your constructive feedback, and we will incorporate these refined examples into the revised manuscript to improve both clarity and motivation.

---

> > > > ### Comment · Reviewer_3grh · 2025-11-24
> > > >
> > > > Thank you for taking the time for finding more relevant examples.
> > > > I have no further questions myself, but I will see how the rebuttal process evolves to make a final decision.

---

> > > > > ### Author Response · Authors · 2025-11-27
> > > > >
> > > > > Thank you once again for generously sharing your time and thoughtful feedback. Your insights have been instrumental in strengthening our manuscript, and we respectfully await your final assessment.

---

### Official Review · Reviewer_pS2q · 2025-11-02

**Soundness:** 2
**Presentation:** 3
**Contribution:** 3
**Rating:** 4
**Confidence:** 2

**Summary:**

The paper introduces COMONet (Convex–Concave and Monotonicity-Constrained Neural Networks), a novel architecture that embeds shape constraints. Unlike prior approaches that handle only subsets of these constraints, COMONet can enforce all eight possible combinations (monotonic increasing/decreasing × convex/concave). The model achieves this using a partially connected modular design with five specialized unit types (convex, concave, monotonic, ReLU, and reflected-ReLU units) that guarantee constraint satisfaction by design. The authors provide theoretical proofs of convexity, concavity, and monotonicity preservation, along with experimental validation on both synthetic and real-world datasets. Empirical results show that COMONet achieves competitive predictive performance while adhering to shape constraints.

**Strengths:**

The paper presents a clear, well-motivated approach that unifies convex, concave, and monotonic constraints under one architecture. Each unit’s properties (convexity, concavity, monotonicity) are formally proven, and the composition of layers preserves these guarantees.

**Weaknesses:**

My biggest concern with the proposed method is that it assumes a fixed, a priori assignment of features to constraint types (e.g., monotonic, convex, concave). This means that each input variable can only satisfy one type of constraint at a time. In practical applications, however, a single feature may exhibit different shape behaviors depending on its interaction with other features (e.g., jointly convex with some, concave with others). The current formulation of COMONet—and its theoretical guarantees in Theorems 4.6–4.8—do not appear to extend to such mixed or conditional relationships. Could the authors clarify whether COMONet can represent or guarantee such cross-constraint interactions?

**Questions:**

See the weakness above.


Minor:
1. Line 137: "monotonixity"
2. Line 146: "represents that set..." -->" represents the set..."
3.  Line 216: "selectively"
4. Line 353: "as shown in the table 5" -->"as shown in table 5"

---

> ### Author Response · Authors · 2025-11-21
>
> **Weaknesses**
>
> We sincerely appreciate your constructive feedback and thoughtful comments. Your observation provides an interesting perspective and highlights a meaningful research direction, especially in more complex systems. To avoid potential ambiguity, we would like to offer clarification regarding the problem setting and the scope of the contributions addressed in this paper.
>
> In line with existing shape-constrained neural networks such as ICNN, SMNN, and SCNN, COMONet adopts global variable-wise shape constraints and enforces them through structural design. In this framework, each input variable is assigned a consistent global constraint type (e.g., convex, concave, or monotonic-decreasing) with respect to the output. This formulation reflects the standard problem setting employed in prior work and defines the intended scope of our method.
>
> To the best of our knowledge, COMONet is the first architecture that jointly supports heterogeneous global constraints—convexity, concavity, and monotonicity—within a single unified model, while guaranteeing each constraint type through structural design. Our contributions should be understood within this established and widely adopted problem definition.
>
> We would also like to emphasize that, we strongly agree that the broader direction suggested by the reviewer is highly valuable for future research. In particular, extending shape constraints to conditional or local/interaction-specific regimes would be a technically challenging yet promising topic. We have reflected this point in the revised discussion/conclusion section and consider it a promising direction to explore in our future work.
>
> In addition, while COMONet focuses on global variable-wise shape constraints, extending the framework toward conditional or interaction-dependent shape behaviors represents a technically challenging yet promising direction.
>
> **Minor**
>
> Thank you very much for pointing out the minor typo. We have corrected it and incorporated the change into the revised version (Blue colored). We sincerely appreciate your careful reading and helpful feedback.
>
>
> We hope that our responses help address your concerns regarding our manuscript. We sincerely appreciate your constructive review.
> — The authors

---

### Meta-Review · Area_Chair_5V6z · 2025-12-31

**Summary:**

Reviewer BzaC discovered that a central theorem in the paper, Thm 4.9, was false, which the authors agreed was the case.  Although the authors have proposed a revision to correct this theorem, for conferences such as ICLR, I believe an incorrect theorem statement in the main section of a paper is a submission is grounds for rejection.

**Reviewer Concerns:**

The authors did provide a revised version of Theorem 4.9 that narrowed the claim.  The lack of reviewer interaction following this correction makes it impossible to have confidence in the correctness of the paper, and regardless, I believe incorrect theorem statements in a submission are grounds fro rejection.  I encourage the authors to carefully re-check all proofs in the appendix and to verify the correctness of their results and then re-submit again -- I think most of the other concerns presented by reviewers were quite minor and addressable.

**Reviewer Scores:**

I think it is likely that all reviewers except BzaC would have decreased their scores, as they did not catch the error in the theorem initially.  I am unsure of how bzac would have changed their score.

---

### Decision · Program_Chairs · 2026-01-26

Reject